# Heavy snowfall event over the Swiss Alps: Did wind shear impact secondary ice production?

Zane Dedekind[*1], Jacopo Grazioli[*2], Philip H. Austin[1], and Ulrike Lohmann[3]

[1]Department of Earth, Ocean, and Atmospheric Sciences, University of British Columbia, Earth Sciences Building, 2207 Main Mall, Vancouver, BC, V6T 1Z4, Canada
[2]Environmental Remote Sensing Laboratory (LTE), École Polytechnique Fédérale de Lausanne (EPFL), Lausanne, Switzerland
[3]Institute of Atmospheric and Climate Science, ETH Zurich, Switzerland
[*]Equally contributing authors

**Correspondence:** Zane Dedekind (zane.dedekind@ubc.ca) and Jacopo Grazioli (jacopo.grazioli@epfl.ch)

**Abstract.** Intense dual-polarization Doppler signatures in conjunction with strong vertical wind shear were observed by an X-band weather radar during a winter time high precipitation event over the Swiss Alps. An enhancement of differential phase shift ($K_{dp} > 1\,^\circ\,km^{-1}$) around $-15\,^\circ C$ suggested that a large population of oblate ice particles was present in the atmosphere. Here, we will show that ice-graupel collisions are a likely origin of this population probably enhanced by turbulence. We perform sensitivity simulations that include ice-graupel collisions of a cold frontal passage to investigate whether these simulations can capture the event better and whether the vertical wind-shear had an impact on the secondary ice production (SIP) rate. The simulations are conducted with the Consortium for Small scale Modeling (COSMO), at a 1 km horizontal grid spacing in the Davos region in Switzerland. The rime-splintering simulations could not reproduce the high ice number concentrations, produced too large ice particles and therefore overestimated the radar reflectivity. The collisional-breakup simulations reproduced both the measured horizontal reflectivity and the ground-based observations of hydrometeor number concentration more accurately ($\sim 20\,L^{-1}$). During 14:30-15:45 UTC the vertical wind shear strengthened by 60% within the region favorable for SIP. Calculation of the mutual information between the SIP rate and vertical wind shear and updraft velocity suggested that the SIP rate is best predicted by the vertical wind shear rather than the updraft velocity. The ice-graupel simulations were insensitive to the size restriction during the conversion from ice to graupel and snow to graupel.

## 1 Introduction

In clouds, ice particles play an important role for Earth's radiation budget and precipitation formation. Precipitation originates predominantly from mixed-phase clouds (MPCs) and ice clouds in the midlatitudes, especially over continental regions (Mülmenstädt et al., 2015; Heymsfield et al., 2015, 2020). The formation of ice particles, therefore, needs to be described adequately if any attempt is made to understand the evolution of MPCs and ice clouds.

Ice formation can occur through primary and secondary ice production (SIP) processes. Primary ice production includes homogeneous freezing of supercooled liquid water at temperatures ($T$) $< \sim-38\,^\circ C$ and heterogeneous ice nucleation of super-

cooled liquid water at warmer subzero $T > \sim-38°C$. After the first formation of ice particles secondary ice processes may occur. In a narrow temperature range, $-3 \geq T \geq -8\,°C$, rime splintering (Hallett and Mossop, 1974) can occur when supercooled liquid water collides with ice particles, freezes from the outside in and shatters as a result of internal pressure buildup.

Rime splintering has been used extensively in models but has been shown to be inadequate, having ice number concentrations orders of magnitude less than observed, to capture SIP in wintertime orographic MPCs (Henneberg et al., 2017; Dedekind et al., 2021; Georgakaki et al., 2022). Ice-ice collisions have been more widely used in models in the last decade (Yano and Phillips, 2011; Phillips et al., 2017; Sullivan et al., 2018; Hoarau et al., 2018; Sotiropoulou et al., 2020; Zhao et al., 2021) since they were first studied in laboratory conditions about four decades ago (Vardiman, 1978; Takahashi et al., 1995). SIP

as a result of ice-ice collisions was shown to contribute significantly to the ice crystal number concentrations and thereby explain the discrepancy between models and observations in the Arctic (Sotiropoulou et al., 2020; Zhao et al., 2021), Antarctic (Sotiropoulou et al., 2021b) and mid-latitudes (Sullivan et al., 2018; Dedekind et al., 2021; Georgakaki et al., 2022). The enhancement of smaller ice particles triggers an increase in the combined growth rates (riming and deposition) of up to 33% resulting in larger latent heat release and stronger updraft velocities (Dedekind et al., 2021). When ice-ice collisions occur in

wintertime orographic MPCs, the general tendency is for riming to decrease. Hence, the deposition growth rate dominates the growth rates of ice particles. Due to the stronger updrafts, ice particles are lofted to higher regions within the cloud reducing the local precipitation rates.

The impact of turbulence associated with baroclinic waves on cloud water and precipitation formation is well known (Baumgartner and Reichel, 1975; Houze and Medina, 2005; Medina and Houze, 2015). Updrafts on the scale of $\sim 10\,km$ from baro-

clinic waves have properties of shear-induced turbulence and it is these small cells of enhanced updraft and turbulence that drive orographic precipitation (Medina and Houze, 2015). In regions associated with mountainous terrain, strong shear layers at low-levels approaching a barrier was emphasized by Houze and Medina (2005) to set up turbulence which in turn aids in precipitation growth (by accretion) on the windward side of a mountain. Medina et al. (2005) showed in idealized simulations that a shear layer can develop as a response of flow to the terrain, by which they concluded that this mechanism, in actual topogra-

phy, caused turbulent overturning which enhanced precipitation formation. In their simulation, the precipitation formation was linked to enhanced accretion (see also Medina and Houze, 2015). The probability of interactions between cloud hydrometeors, whether through riming and/or aggregation, increases with turbulence and aids in the rapid formation of precipitation regardless of whether the turbulence is associated with orographic flow regimes or in warm conveyor belts (Houze and Medina, 2005; Gehring et al., 2020). These interactions are not limited to the accretion of cloud hydrometeors which causes them to grow but

instead could also cause the fracturing of ice particles in ice-ice collisions enhancing SIP. Dedekind et al. (2021) hypothesized that ice-graupel collisions could also be sensitive to the rate at which graupel forms, which is a function of the size of ice particle and the riming rate. In the Seifert and Beheng (SB 2006) two-moment cloud microphysics scheme (2M), ice crystals or snow undergoing riming can only be converted to graupel once they reach a size of $200\,\mu m$. Remote sensing from weather radars is used to study snowfall microphysics and hydrometeors' habit (e.g., shape, phase or hydrometeor type). Although radar

observations do not provide direct information on SIP, a few studies leveraged the Doppler and/or dual-polarization capabilities of weather radars to identify the occurrence of SIP and to speculate case-by-case on the possible mechanisms behind its origin.

Two non mutually exclusive approaches can be found in the literature. Zawadzki et al. (2001); Oue et al. (2015); Luke et al. (2021) exploited Doppler spectra collected by vertically-pointing radars to identify the appearance of secondary populations of particles at given altitudes or temperature levels. Other approaches (Hogan et al., 2002; Andrić et al., 2013; Sinclair et al., 2016; Kumjian and Lombardo, 2017) focused on the interpretation of the signature of dual-polarization variables and their respective evolution over the vertical column of precipitation. This second approach, also used in this study, leverages the fact that dual-polarization variables are complementary and impacted differently by changes in number, shape, size and density of hydrometeors. Additional information (in-situ, models, or a combination of more radars) is typically needed to increase confidence in the retrievals collected.

The principle of dual-polarization for weather radars relies on radars transmitting pulsed horizontally and vertically polarized waves (Field et al., 2016). The waves interact with precipitation and, looking at the differences in power and phase of the echoes in each polarization, information about the orientation, size, number concentration (and phase) of the hydrometeors being sampled can be retrieved. Horizontal (vertical) reflectivity $Z_H$ ($Z_V$) and differential reflectivity $Z_{DR}$ are variables that exploit the power intensity of the echoes. $Z_H$ [dBz] increases as particles get larger and/or denser and/or more numerous. $Z_{DR}$ [dB] is the difference $Z_H - Z_V$ and can be used to distinguish oblate particles ($Z_{DR} > 0$) from prolate ones ($Z_{DR} < 0$), while it has near-zero values for spherical particles. In an environment where preferentially-oriented anisotropic ice particles are dominant, $Z_{DR}$ deviations from near-zero values are frequently observed (Bader et al., 1987; Kumjian et al., 2014). When ice particles form aggregates, and become larger and less oblate, $Z_{DR}$ decreases while $Z_H$ increases (Schneebeli et al., 2013; Kumjian et al., 2014; Grazioli et al., 2015a). The backscattered power is different for horizontal and vertical polarizations in the presence of anisotropic particles, as is the propagation speed of the waves. The rate of change in phase shift between the horizontal and vertical polarized echoes is expressed by the specific differential phase shift $K_{dp}$ [°km$^{-1}$]. This variable is complementary and not redundant; it is in fact not affected by the absolute calibration of a radar and is less affected than $Z_{DR}$ by eventual presence of large isotropic particles within the sampling volume. For instance, local $K_{dp}$ enhancements during snowfall have been documented (Schneebeli et al., 2013; Bechini et al., 2013) and in some cases been associated with SIP (e.g., Andrić et al., 2013; Grazioli et al., 2015a; Sinclair et al., 2016). Grazioli et al. (2015a) suggested in a case study that an increase of $K_{dp}$ can be due to a very large number concentrations of rimed anisotropic ice crystals resulting from ice-ice collision . A recent study (von Terzi et al., 2022) suggested that the $K_{dp}$ enhancement was due to a combination of secondary ice production and an appropriate temperature range $T \approx -15°C$ (where growth of planar crystals by vapor deposition, dendrites in particular, is maximized) can lead to this signature. Dendrites have very low densities, favor aggregation below (hence the increase of $Z_H$ below $K_{dp}$ peaks) and can easily fracture on impact with other ice particles.

In this paper, we propose that the vertical wind shear associated with a cold-front passage over the mountains of eastern Switzerland enhanced the formation of small and numerous oblate ice particles through ice-ice collisions. The ice-ice collisions explain the peculiar signatures in the data collected by a Doppler dual-polarization radar deployed in the region.. We address the following questions:

– Can these radar signatures be attributed to high ice number concentrations linked to SIP other than rime splintering?

– By including ice-graupel collisions in the model, can we simulate the high ice number concentrations that were observed?

– Was there a correlation between the vertical wind shear and SIP?

– How sensitive are SIP rates to the conversion rate of ice particles to graupel?

## 2 Methods

### 2.1 Weather radar and 2 dimensional video disdrometer (2DVD)

A X-band dual-polarization mobile Doppler weather radar (MXPol) of the École Polytechnique Fédérale de Lausanne Environmental Remote Sensing Laboratory (EPFL-LTE) was set up at 2 133 m a.s.l. on a ski slope overseeing the valley of Davos (Schneebeli et al., 2013) from the southern side as shown in figure 1. Its exact location was 46.789° N and 9.843° E (see Sec 2.1). MXPol is well suited for deployment in complex Alpine terrains or remote locations (e.g. Schneebeli et al., 2013; Grazioli et al., 2015a, 2017) and operated from September 2009 to July 2011. The radar was routinely scanning over the valley of Davos in a sequence including pseudo-horizontal scans (fixed elevation and variable azimuth) and 2D vertical cross-sections (fixed azimuth scans with elevation ranging from $0\,°$ to $90\,°$, better known as Range Height Indicator or RHI scans). One RHI scan in particular, used as a data source of this study, was conducted every 5 minutes towards NE, at an azimuth of $22\,°$. Only observations collected at elevation angles below $40°$ are used, in order to limit the effect of elevation dependencies on the polarimetric variables (Ryzhkov et al., 2005). MXPol provides single ($Z_{\mathrm{H}}$) and dual-polarization ($Z_{\mathrm{DR}}$, $K_{\mathrm{dp}}$, $\rho_{\mathrm{HV}}$) measurements as well as Doppler data which have been proven useful in several snowfall microphysics studies (e.g. Schneebeli et al., 2013; Grazioli et al., 2015a; Kumjian and Lombardo, 2017; Oue et al., 2021). Additionally, retrieval algorithms adapted to polarimetric data allow one to estimate properties such as hydrometeor type (Grazioli et al., 2015b, as used in this work) or, under given assumptions, microphysical quantities such as ice number concentration $N_{\mathrm{t}}$, median volume diameter $\bar{D}_{\mathrm{m}}$ or ice water content (IWC). The hydrometeor classification method discriminates between three ice-phase dominant hydrometeor types: individual ice crystals, aggregates and rimed particles.

The microphysical quantities can instead be estimated from combinations of $Z_{\mathrm{H}}$, $Z_{\mathrm{DR}}$, $K_{\mathrm{dp}}$ and the radar wavelength following Murphy et al. (2020):

$$IWC = 4 \times 10^{-3} \frac{K_{\mathrm{dp}} \lambda}{1 - Z_{\mathrm{dr}}^{-1}} \tag{1}$$

$$log_{10}(N_{\mathrm{t}}) = 0.1 Z_H - 2 \log_{10}\left(\frac{Z_{\mathrm{dp}}}{K_{\mathrm{dp}} \lambda}\right) - 1.11 \tag{2}$$

$$\bar{D}_{\mathrm{m}} = -0.1 + 2\left(\frac{Z_{\mathrm{dp}}}{K_{\mathrm{dp}} \lambda}\right)^{0.5} \tag{3}$$

In these equations, IWC is expressed in $[\mathrm{g\,m^{-3}}]$, $N_t$ in $[\mathrm{L^{-1}}]$ and $\bar{D}_m$ in $[\mathrm{mm}]$. $\lambda$ is the radar wavelength in $[\mathrm{mm}]$, $Z_{dr} = 10^{0.1\,Z_{DR}}$ is the differential reflectivity in linear units, $Z_{dp} = 10^{0.1\,Z_H} - 10^{0.1\,Z_V}$ is the reflectivity difference in linear units $[\mathrm{mm^6\,m^{-3}}]$. More details about the derivation of the equation can be found in Ryzhkov and Zrnic (2019); Murphy et al. (2020) but it is worth to focus on the main assumptions and limitations. The main assumptions are:

- The equations are derived assuming to be in the Rayleigh regime (which may not be fulfilled) for the X-band radar for large hydrometeors.

- The density and the size of the hydrometeors are assumed to be inversely proportional.

The retrievals have shown to be most reliable at $T < -10\,^{\circ}\mathrm{C}$, for low riming degrees and in regions where the $K_{dp}$ and $Z_{DR}$ signals are not close to 0. As recognized by Murphy et al. (2020), the errors may be large and in situ validation efforts are needed to refine these techniques. As a final caveat, the equations developed on theoretical basis are in practice very sensitive to the accuracy of the polarimetric variables, which can be very noisy. $K_{dp}$ in particular is an estimated variable affected by mean errors on the order of 30% (Grazioli et al., 2014a).

An additional ground-based source of information for this event is provided by a 2 dimensional video disdrometer, 2DVD (For more information about this instrument at this location see Grazioli et al., 2014b) which was deployed on the opposite side of the Davos valley with respect to MXPol (46.830° N and 9.810° E, 2543 m. amsl). The 2DVD measures the size and fall velocity of hydrometeors captured within its measurement area of $11\,\mathrm{cm} \times 11\,\mathrm{cm}$. The 2DVD is used in this study as ground reference to quantify the number concentration of snowfall particles (larger than 0.2 mm in terms of maximum dimension, according to the sensitivity of the instrument itself).

## 2.2 Model setup

### 2.2.1 Spatial and temporal resolution

The Consortium for Small Scale Modelling (COSMO; Baldauf et al., 2011) non-hydrostatic model, version 5.4.1b, was used for this case study. COSMO has been used to study wintertime (Lohmann et al., 2016; Henneberg et al., 2017; Dedekind et al., 2021) and summertime (Dedekind, 2021; Eirund et al., 2021) orographic MPCs in the Swiss Alps. The model domain roughly covers a region of $500\,\mathrm{km} \times 600\,\mathrm{km}$ (44.5 to 49.5° N and 4 to 13° E) at a horizontal grid spacing of $1.1\,\mathrm{km} \times 1.1\,\mathrm{km}$ (Fig. 1). A height based hybrid smoothed level vertical coordinate system (Schär et al., 2002) with 80 levels is used and stretched from the surface to 22 km. For this study, we simulate the cold front passage between 11:00 and 18:00 UTC and analyze the results between 13:00 and 18:00 UTC on March 26, 2010. COSMO is forced with hourly initial and boundary conditions re-analysis data at a horizontal resolution of $7\,\mathrm{km} \times 7\,\mathrm{km}$, supplied by MeteoSwiss. The model time step is 4 s with an output frequency every 15 min.

Simulations were conducted including several SIP processes, which consisted of ice-graupel collisions (as thoroughly discussed in section 2.2.2 below) and a control simulation, referred to as the rime splintering (RS) simulation, where only rime splintering was active. For each of these setups, 5 ensemble simulations are conducted by perturbing the initial temperature

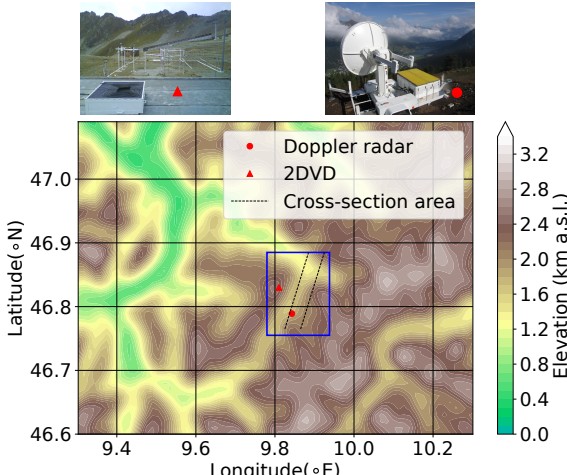

**Figure 1.** Overview of the model orography and the instrument location setup. The parallelogram (dashed black lines) is the domain of the flow-oriented vertical cross-section analysis in section 3.1 following the direction of the dual-polarized Doppler radar MXPol (the red dot located at 46.789° N and 9.843° E) data. The blue box is the domain used for analysis in section 3.2 and 3.3. The red triangle is the location of the ground-based video disdrometer.

conditions at each grid point through the model domain with unbiased Gaussian noise at a zero mean and a standard deviation of 0.01 °C (Selz and Craig, 2015; Keil et al., 2019). The model output was interpolated along the mean of three vertical cross-sectional paths, $\approx 3\,\mathrm{km}$ wide, similar to direction of the RHI cross section conducted by the weather radar (Fig. 1). The cross-sections of the simulations were averaged to create a mean cross-section which is then compared to the radar data. To generate the Hovmöller diagrams, we further took the mean along the length of the cross-section for both the simulations and the radar data.

### 2.2.2 Cloud microphysics scheme

We use a detailed two-moment bulk cloud microphysics scheme within COSMO with six hydrometeor categories, including cloud droplets, rain, ice, snow, graupel and hail (Seifert and Beheng, 2006). The 2M scheme has been used extensively to study the evolution, lifetime, persistence and aerosol-cloud interactions of MPCs (Seifert et al., 2006; Baldauf et al., 2011; Lohmann et al., 2016; Possner et al., 2017; Henneberg, 2017; Glassmeier and Lohmann, 2018; Sullivan et al., 2018; Eirund et al., 2019, 2021). We refer to ice particles as any combination of the hail, graupel, snow or ice categories. Cloud droplet activation is based on an empirical activation spectrum which depends on the cloud-base vertical velocity and the prescribed number concentration of cloud condensation nuclei (Seifert and Beheng, 2006). The application is appropriate in atmospheric models with a horizontal grid size and time resolution of $\Delta x \leq 1\,\mathrm{km}$ and $\Delta t < 10\,\mathrm{s}$ respectively. The warm-phase autoconversion process from Seifert and Beheng (2001) was updated with the collision efficiencies from Pinsky et al. (2001) and also takes into account the decrease in terminal fall velocity associated with an increase in air density. A better approximation of the collision

rate between hydrometeors was also introduced by Seifert and Beheng (2006), which makes use of the Wisner-approximation (Wisner et al., 1972).

The primary production of ice formation is described by the homogeneous and heterogeneous nucleation pathways. Homogeneous freezing of cloud droplets, parameterized from the homogeneous freezing rates of Cotton and Field (2002), is calculated for $0 > T \geq -50\,°C$. At $-38\,°C$ most cloud droplets will freeze given the enhanced homogeneous nucleation rates at colder temperatures. As a lower bound the homogeneous freezing of all cloud droplets occurs at $T = -50\,°C$. The homogeneous nucleation of solution droplets, typically associated with cirrus cloud formation follows Kärcher et al. (2006). Here, the number density and size of nucleated ice crystals is determined by the vertical wind speed, temperature and pre-existing cloud ice. Heterogeneous nucleation is empirically derived which depends on the chemical composition and surface area of multiple species of aerosols, namely organics, soot and dust (Phillips et al., 2008).

Secondary ice production through rime splintering is the only process that is included in the standard version of COSMO which has been used extensively in other numerical weather models (Blyth and Latham, 1997; Ovtchinnikov and Kogan, 2000; Phillips et al., 2006; Milbrandt and Morrison, 2016; Phillips et al., 2017). In COSMO, rime splintering occurs at $-3 \geq T \geq -8\,°C$ (Hallett and Mossop, 1974) when supercooled droplets and rain drops ($D_{c,r} \geq 25\,\mu m$) collide with ice hydrometeors ($D_{i,s,g} \geq 100\,\mu m$) (e.g Seifert and Beheng, 2006). A default value of 350 fragments per milligram of rime is used in the rime splintering parameterization. Another SIP process, collisional breakup, was added to COSMO and tested in several studies (Sullivan et al., 2018; Dedekind et al., 2021). During collisional breakup graupel collides with either ice and/or snow particles and fractures. This can increase the ice particles at temperatures warmer than $-21\,°C$. Several studies (e.g Sotiropoulou et al., 2021b; Dedekind et al., 2021) have considered reducing the effectiveness of the collisional breakup parameterization due to the discrepancies in the large hail particles and their corresponding fall velocities used in the laboratory study conducted by Takahashi et al. (1995). Because the parameterization from Sullivan et al. (2018) is based on experimental results by Takahashi et al. (1995), it is constrained to only ice-graupel collisions and may not be adequate when studying ice-ice collisions in wintertime MPCs consisting of mainly ice and snow crystals. The amount of fractures that can be generated in snow-ice collisions might, on the contrary, not be significant because of the low collision kinetic energy between unrimed particles. Experimental studies are thus required to show evidence for generated ice fractures between unrimed ice particles. In this study of the heavy snowfall event during which high $K_{dp}$ values were recoreded, we use the parameterizations for ice-graupel collisional breakup (BR) from Dedekind et al. (2021, BR28 and BR2.8T) and Sotiropoulou et al. (2021b, BR-Sot) in COSMO in different forms:

$$\aleph_{BR} = \frac{F_{BR}}{\alpha}(T - 252)^{1.2}\exp\left[-(T - 252)/\gamma_{BR}\right], \quad \text{for BR28} \quad (F_{BR}, \alpha, \gamma_{BR}) = (280, 10, 2.5) \tag{4}$$

$$\aleph_{BR} = \frac{F_{BR}}{\alpha}(T - 252)^{1.2}\exp\left[-(T - 252)/\gamma_{BR}\right], \quad \text{for BR2.8T} \quad (F_{BR}, \alpha, \gamma_{BR}) = (280, 100, 5) \tag{5}$$

$$\aleph_{BR} = F_{BR}(T - 252)^{1.2}\exp\left[-(T - 252/\gamma_{BR})\right]\frac{\bar{D}}{\bar{D}_0}, \quad \text{for BR-Sot} \quad (F_{BR}, \bar{D}_0, \gamma_{BR}) = (280, 0.02, 5) \tag{6}$$

where $\aleph_{BR}$ is the number of fragments generated per collision, $\alpha$ is the scale factor, $F_{BR}$ is the leading coefficient, $T$ is the temperature in Kelvin, $\gamma_{BR}$ is the decay rate of the fragment number at warmer temperatures, $\bar{D}$ is the diameter of particle undergoing fracturing and $\bar{D}_0$ is the diameter of the hail particles used in Takahashi et al. (1995). Only using ice-graupel collisions would limit the full description of SIP as a result of wind shear when graupel formation becomes restricted. Because of the above mentioned inconsistency between the hail particles and their corresponding fall velocity used in Takahashi et al. (1995), which is described in more detail in Dedekind et al. (2021), all the parameterizations (Eqs. 4, 5 and 6) have scaling factors. Equations 4 and 5 were applied in Dedekind et al. (2021) for the BR28 and BR2.8T simulations, respectively. Equation 4 is scaled by $\alpha = 10$ and has a slower decay rate of fragment number at warmer temperatures represented by $\gamma_{BR} = 2.5$ and equation 5 is scaled by 100 while using the same decay rate of fragment numbers of $\gamma_{BR} = 5$ as used in Sullivan et al. (2018) which was derived from Takahashi et al. (1995). Equation 6, for the BR-Sot simulation, was applied in Sotiropoulou et al. (2021b). They used a scaling parameter, $\bar{D}/\bar{D}_0$, that was applied to the breakup parameterization from Sullivan et al. (2018) where $\bar{D}_0 = 0.02$ m. Similar to Dedekind (2021), the ICNC in COSMO is limited to $2\,000\,\mathrm{L}^{-1}$. Furthermore, Dedekind (2021) concluded that the conversion rate from ice crystals or snow to graupel, which is a function of the riming rate of ice crystals or snow with raindrops, may contribute to enhanced collisional breakup (Seifert et al., 2006). In eq. (70) of Seifert and Beheng (2006), they specify that ice and snow crystals can only be converted to graupel once they reach $\bar{D}_{i,s} \geq 500\,\mu m$. However, in the current version of the 2M scheme (as used in this study), ice and snow crystals are converted to graupel already once they exceed $\bar{D}_{i,s} \geq 200\,\mu m$. Therefore, earlier graupel formation is promoted in the current version which should lead to enhanced SIP though ice-graupel collisions. To test the model's sensitivity to these different thresholds for graupel formation, we set-up sensitivity studies with graupel formation at $\bar{D}_{i,s} \geq 300, 400$ and $500\,\mu m$, respectively, to understand how the conversion rate impacts SIP processes. To accomplish this we change the ice category conversion size requirement, $\bar{D}_{i,s}$, during riming from $200\,\mu m$ (BR2.8T), to $300\,\mu m$ (BR2.8T_300), to $400\,\mu m$ (BR2.8T_400) and lastly to $500\,\mu m$ (BR2.8T_500).

To investigate the impact of vertical wind shear and updraft on SIP the probability density functions (PDFs) for the variables from the collisional breakup simulations are analyzed. Furthermore, the joint PDFs are calculated along with the mutual information (MI, Shannon and Weaver, 1949) score which quantifies the strengths of dependencies between the SIP rate and cloud properties (e.g., Dawe and Austin, 2013). For this purpose a $10\,\mathrm{km} \times 10\,\mathrm{km}$ region was selected and masked by the levels in which SIP occurred ($T > -21\,^\circ C$) from 15:15 to 16:30 UTC. This resulted in the 16 121 data points for which an expression from Hacine-Gharbi et al. (2013) was used for finding the optimal number of bins (17 bins in our case) to estimate the MI for continuous random variables.

## 3 Results

### 3.1 The case study

A synoptic system passed over Switzerland on 26 March 2010. The cold front was associated with a south-westerly wind flow at higher altitudes, the development of peculiar polarimetric radar signatures (e.g. an intense $K_{dp}$), vertical wind shear closer to the surface below 4 km amsl, a surface temperature drop of $\sim 7\,^\circ C$ (Fig. S1) and intense snowfall in the afternoon. In particular

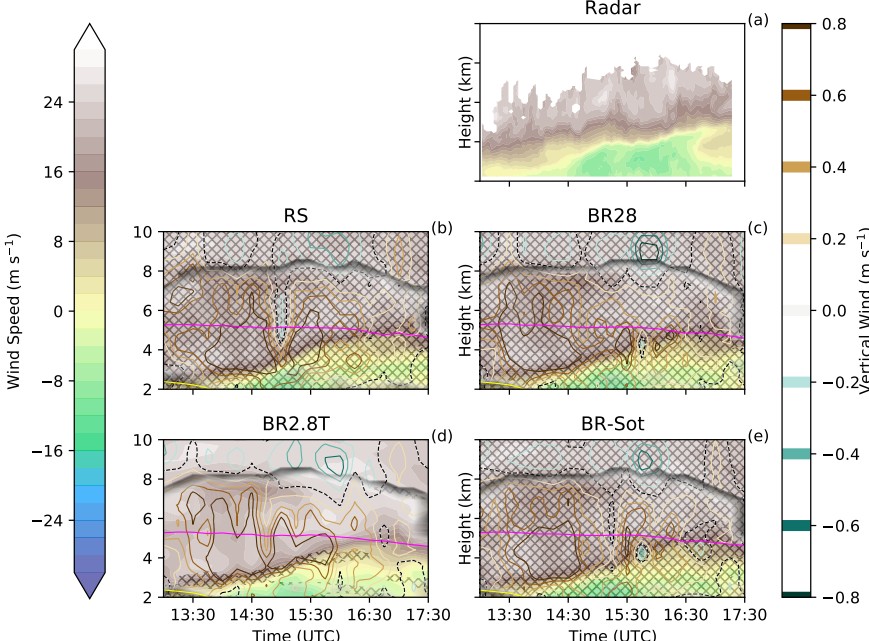

**Figure 2.** Hovmöller diagrams of wind speed and vertical velocity for panels **(a)** Doppler radar, **(b)** RS , **(c)** BR28, **(d)** BR2.8T and **(e)** BR-Sot between 13:00 and 17:30 UTC. The blue and red colors denote wind blowing towards and away from the radar respectively. The pink line is the $-21\,°$C isotherm. At warmer temperatures collisional breakup occurs. The shaded gray area is the cloud area fraction. The hatched is the region where the air layer is dynamically unstable determined by a bulk Richardson number of less than 0.25.

$K_{dp}$ reached values around $1.5\,°\,\mathrm{km}^{-1}$ at certain height levels and towards the end of the event it was exceeding $2\,°\,\mathrm{km}^{-1}$ (Fig. 3a). $K_{dp}$ scales with radar frequency. A statistical analysis of $K_{dp}$ in snowfall conducted with this radar and in this location

over a long observation period (Schneebeli et al., 2013), showed that the 80% quantile of $K_{dp}$ at every height level is lower than $0.5\,°\,\mathrm{km}^{-1}$. Considering that the distribution of $K_{dp}$ is very skewed, values above $1\,°\,\mathrm{km}^{-1}$ in snow can be considered as unusually large.

Wind shear, observed by the dual-polarization Doppler radar, was visible between 2 and 5 km amsl (Fig. 2a). The wind velocity at lower altitudes shifted from southerly to a northerly which was captured in all simulations, albeit not as prominent

as the observations (Figs. 2b-e). The wind shear and the associated updrafts may have contributed to an enhanced SIP rate between 3 and 5 km in the collisional breakup simulations (Fig. 2b-e). Here, the bulk Richardson number, which is a ratio of the buoyant energy to shear-kinetic energy, is determined to assess the dynamic stability of the air layer. An air layer becomes turbulent if the Richardson number is less than the critical Richardson number of 0.25 (e.g., **?**). Figure 2b-e show where the air layer was turbulent with enhanced interactions between ice particles and could have caused enhanced ice-graupel collisions.

Regions of enhanced updraft, where hydrometeors can grow to larger sizes, were mostly seen immediately above the turbulent layer.

The sub-zero temperature in the region of enhanced $K_{dp}$ was warmer than $-21\,°C$, which is in the temperature range favourable for ice-graupel collisions (Takahashi et al., 1995). We thus hypothesize that the in-situ cloud conditions together with the vertical wind shear could have triggered higher secondary ice production rates that can be reflected in radar mea-
surements, as $K_{dp}$ is only an indicator of number high concentrations of oblate hydrometeors in the radar sampling volume (Kennedy and Rutledge, 2011; Bechini et al., 2013; Grazioli et al., 2015a; von Terzi et al., 2022).

## 3.2 Simulated vs observed radar reflectivity

### 3.2.1 Model and Doppler radar comparison

Horizontal reflectivity $Z_H$ is used to compare the model to the observations throughout the cloud and to analyze the impact of
secondary ice production on the simulated radar reflectivity. During the early afternoon, the median of $Z_H$ remained mostly below 20 dBz. At around 15:15 UTC larger ice hydrometeors were present (either as a result of enhanced aggregation or depositional growth) between 4 and 6 km amsl which then started to sediment (Fig. 3c and d). A peak in $Z_H$ at 3 km amsl was observed in the fall streaks when the cloud droplets rimed onto the sedimenting ice hydrometeors. The radar-derived hydrometeor classification showed that much of the ice hydrometeor growth occurred through aggregation and riming. At
15:30 UTC, a very high median $K_{dp} > 1\,°\,km^{-1}$ and $Z_{DR} > 1\,dB$ were observed (Fig. 7e). The vertical evolution of $K_{dp}$ and $Z_{DR}$ is similar, with a peak observed about 4 km amsl, which is 1 km above the peak in $Z_H$ (Fig. 7d). The large and colocated values of $Z_{DR}$ and $K_{dp}$ suggest that a large population of oblate particles were present at these heights. The increase in $Z_H$ and colocated decrease of $K_{dp}$ and $Z_{DR}$ below suggest that larger (and/or denser) and more isotropic particles were forming. Aggregation and riming, not mutually exclusive, both explain this behavior.
During the entire event (Fig. 3) the peak of $Z_{DR}$ was more often above the peak of $K_{dp}$, suggesting that the population of particles in the areas of enhanced $K_{dp}$ also included larger isotropic aggregates. The occurrence of peaks in polarimetric variables at certain heights above ground ($K_{dp}$ in particular) has been observed during intense snowfall events (e.g. Kennedy and Rutledge, 2011; Schneebeli et al., 2013; Grazioli et al., 2015a). The $K_{dp}$ enhancement in particular has often been observed near the $-15°C$ isotherm and has been interpreted as the signature of enhanced dendritic growth (Kennedy and Rutledge, 2011;
Bechini et al., 2013) in combination with secondary ice production (von Terzi et al., 2022). Dendrites are prone to aggregation and therefore the $K_{dp}$ peak disappears (and $Z_H$ increases) as particles approach the ground level.

$Z_H$ was significantly overestimated by the RS simulation between 13:00 and 17:30 UTC which most likely was a result of the following chain of events. 1) Insufficient droplets of size 25 μm (Fig. 5d), within the narrow temperature range ($-3 \geq T \geq -8\,°C$), led to a limitation in ice particle growth by riming and therefore limited rime splintering (Fig. ??f). 2) Be-
cause rime splintering was not that active, typical for wintertime MPCs (e.g., Henneberg et al., 2017; Dedekind et al., 2021), the ice and snow crystals grew mainly by depositional growth and aggregation. 3) The ice and snow crystal size distributions widened substantially (Figs. 6a, b and S2a, b). These categories both had number concentrations less than $100\,L^{-1}$ with particle diameters of up to 0.8 and 5.1 mm, respectively, at 15:30 UTC. 4) The larger ice and snow crystal diameters resulted in enhanced $Z_H$. These observation is consistent with other times during the day which showed even larger sized ice and snow

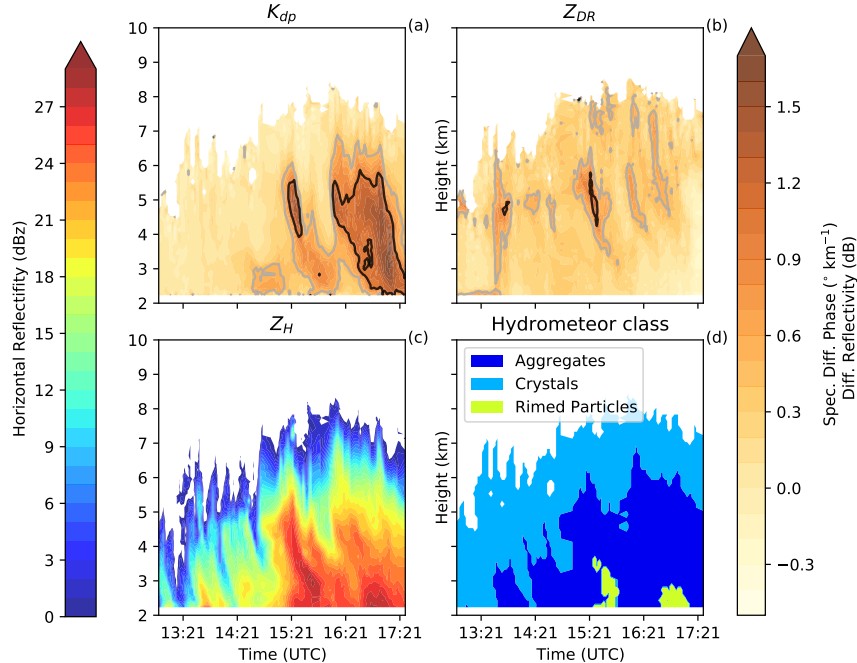

**Figure 3.** Hovmöller diagrams of the **(a)** spectral differential phase ($K_{\mathrm{dp}}$), the **(b)** differential reflectivity ($Z_{\mathrm{DR}}$), **(c)** the horizontal reflectivity ($Z_{\mathrm{H}}$) and **(d)** the hydrometeor class categories derived from the Doppler radar between 13:00 and 17:30 UTC. The grey and black lines in panels **(a and b)** are where both $K_{\mathrm{dp}}$ and $Z_{\mathrm{DR}}$ are larger than 0.5 and 1, respectively.

crystals of 0.9 and 5.2 mm, respectively (Figs. S3a and S4a) as well as higher rain mass mixing ratios (e.g. Fig. 5a). There were single grid points where snow crystal even reached diameters of 13 to 17 mm during the latter part of the day (not shown here). Additionally, excessive size sorting in the model most likely contributed to the overestimation in $Z_{\mathrm{H}}$. Size sorting typically occurs within the sedimentation parameterization of 2M schemes in regions of vertical wind shear or updraft cores (Milbrandt and McTaggart-Cowan, 2010; Kumjian and Ryzhkov, 2012). All these factors contributed to the RS simulation overestimating

$Z_{\mathrm{H}}$ by at least 8 dBz throughout the vertical profile compared to the observations (Fig. 7d).

When collisional breakup was allowed to occur in the BR28, BR2.8T and BR-Sot simulations, the ice particles from the ice and snow crystals did not have time to grow as large compared to the RS simulation. Throughout the vertical profile below 6 km at 15:30 UTC, the ice number concentration was at least an order of magnitude larger than expected from the RS simulation with a SIP rate in excess of $20\,\mathrm{L}^{-1}\mathrm{s}^{-1}$ (Fig. 7c, f). In both figures 7 and S5 the observed ice crystal number concentration

recorded by the disdrometer was remarkably well represented at the surface by the BR2.8T and BR-Sot simulations (similar results are shown in Dedekind et al., 2021). The ice crystal and snow number concentrations were orders of magnitudes larger for $\bar{D}_{\mathrm{i}} < 0.4$ mm and $\bar{D}_{\mathrm{s}} < 0.8$ mm respectively compared to the RS simulation (Fig. 6a and b). The smaller ice particles caused a reduction in $Z_{\mathrm{H}}$ which compared better to the observations than for the RS simulations. It is also likely that the narrower

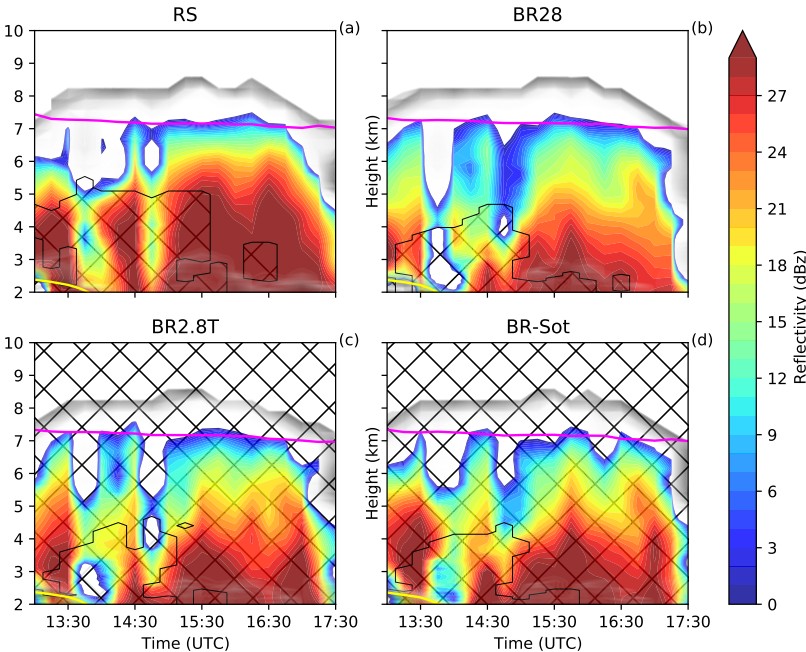

**Figure 4.** Hovmöller diagrams of the simulated reflectively for panels **(a)** RS , **(b)** BR28, **(c)** BR2.8T and **(d)** BR-Sot between 13:00 and 17:30 UTC. The hatched area is defined as the MPC where the cloud droplet mass concentration and ice mass concentration is greater than 10 and 0.1 mg m$^{-3}$ respectively. The pink line is the homogeneous freezing line at $-38\,°C$, and the shaded gray area is the cloud area fraction.

ice crystal and snow size distributions meant that the excessive size sorting in 2M schemes may have not been as pronounced,
which contributed to the lowering of $Z_H$.

During the late stage of the snowfall event, at 17:00 UTC, the replenishment of graupel diminished rapidly (Figs. 5 and S5c) causing a substantial reduction in the SIP rate (Fig. S5). Less collisional breakup allowed the ice and snow crystals to grow to larger sizes, $\bar{D}_i \sim 1.2$ mm and $\bar{D}_s \sim 3.3$ mm respectively, primarily through deposition and/or aggregation (Fig. S2a, b). A lower $Z_{DR}$ is consistent with less anisotropic particles produced by aggregation and/or riming. However, the enhanced
concentration of oblate particles (increase in $K_{dp}$) was in contrast to the simulations showing a reduction in cloud content as the cloud began to dissipate earlier than in the observations. None of the simulations were able to describe the high ice particle formation event that was most likely triggered through ice-ice collisions of dendrites given the favorable temperature range. In the event that snow (e.g., ice-snow collisions) would have also been considered as a collider species in the simulations (e.g., Sotiropoulou et al., 2021a), they might have been able to simulate higher ice particle formation rates as suggested in the high
$K_{dp}$ radar observations. The breakup simulations, in general, did simulate $Z_H$ more accurately than the conventionally used rime-splintering scheme and did show to improve the ice crystal number concentration at the surface and in the vertical column.

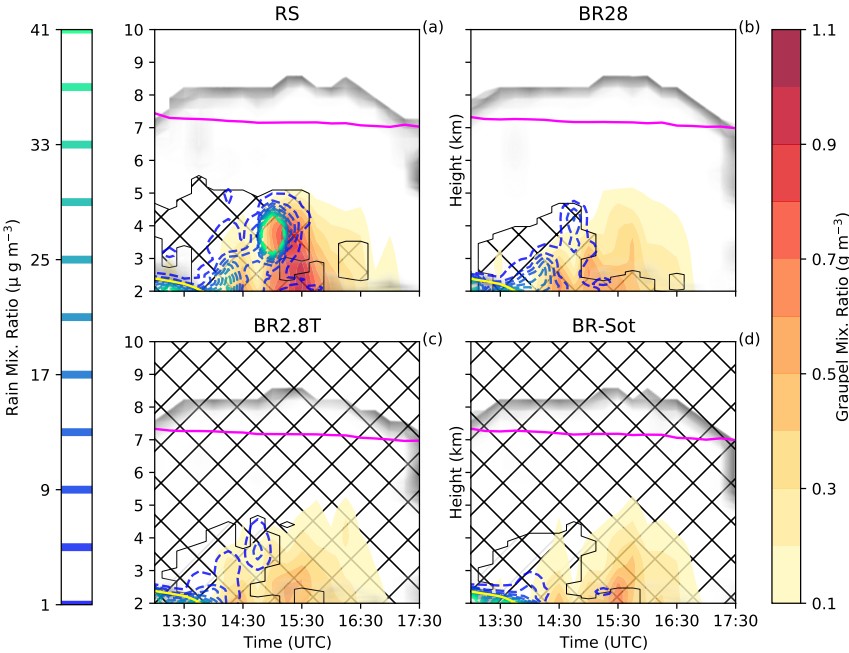

**Figure 5.** Hovmöller diagrams of graupel and rain mixing ratio for panels **(a)** RS , **(b)** BR28, **(c)** BR2.8T and **(d)** BR-Sot between 13:00 and 17:30 UTC. The hatched area is defined as the MPC where the cloud droplet mass concentration and ice mass concentration is greater than 10 and 0.1 mg m$^{-3}$ respectively. The pink line is the homogeneous freezing line at $-38\,^{\circ}$C, and the shaded gray area is the cloud area fraction.

### 3.2.2 Differences in collisional breakup simulations

The parameterization for BR28 (Eq. 4) was set up such that more (less) ice fractures are generated at colder (warmer) temperatures than $-10\,^{\circ}$C compared to the BR2.8T simulation (Eq. 5). Although the graupel number concentration, which is responsible for generating ice fractures upon colliding with other ice particles decreased with altitude (Fig. 5b,c and d), the BR28 simulation still generated 8 times more ice particles than the BR2.8T and BR-Sot simulations at 4 km amsl at $T \sim -15\,^{\circ}$C (Fig. 7f). At temperatures of $-10\,^{\circ}$C (3 km amsl), the SIP rate decreased rapidly from 100 to almost $0\,\mathrm{L}^{-1}\mathrm{s}^{-1}$ at the surface. As a result of the lower SIP rates (less ice-graupel collisions) compared to the BR2.8T and BR-Sot simulations, there were several implications: 1) ice crystals and snow particles had more time to grow to larger sizes as seen in the wider particle size distributions (Fig. 6); 2) the number of ice hydrometeors was an order of magnitude below (worst in the collisional-breakup simulations) the observed ground-based video disdrometer observations of $\sim 20\,\mathrm{L}^{-1}$ (for particles less than 0.2 mm) at 15:30 and 17:00 UTC (Figs. 7c and S5c) and; 3) interestingly, the simulated $Z_{\mathrm{H}}$ compared better with the radar observations although the ice hydrometeors were underestimated (Figs. 7d and S5d). It is not clear which one of the collisional-breakup simulations performed better in general because one performs better against one set of observations and then worse against another set of

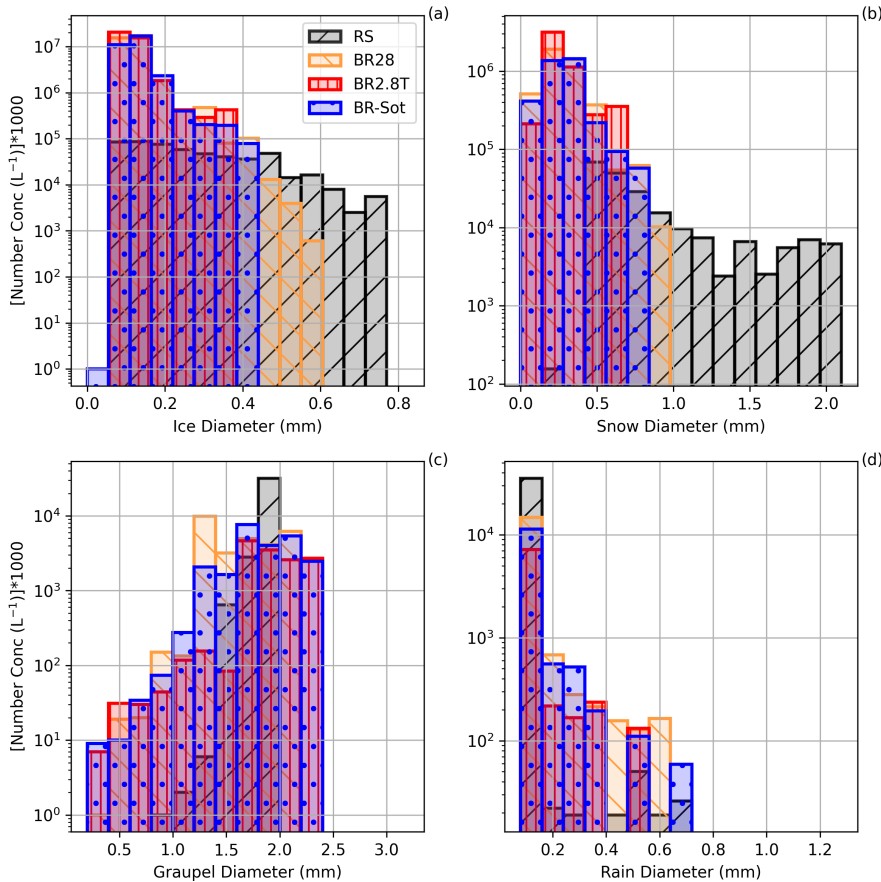

**Figure 6.** Particle size distribution over the number concentrations for panels **(a)** ice, **(b)** snow, **(c)** graupel and **(d)** raindrops for all the simulations at 15:30 UTC.

observations. However, ice-graupel collisions are important for representing the observed $Z_H$ better. For instance, the breakup of fast-growing, low density, dendrites at $T \sim -15\,°C$ reduces the ice particle size and therefore reduces the simulated $Z_H$. Figure S5(d and f) showed a spike in the SIP rate at these temperatures resulting in a better agreement with the observed $Z_H$.

### 3.3 Process understanding

In this section, the dependence of the SIP rate on wind patterns is examined over the region (blue box) depicted in figure 1 during three time periods; 13:00-14:15 UTC (early: Fig. S6), 14:30-15:45 UTC (middle: Fig. 8) and 16:00-17:15 UTC (late: Fig. S7). The early period was categorized with strong wind shear and a turbulent layer below 3 km (Fig. 2a-e). During the middle period, the turbulent layer extended to 4 km during which graupel increased in the MPC. The late period was categorized by less wind shear causing the dissipation of the turbulent layer. The cloud entered a glaciated state during this time. We analyzed the middle period because it was the most important period, according to the model, in terms of SIP and is therefore shown in

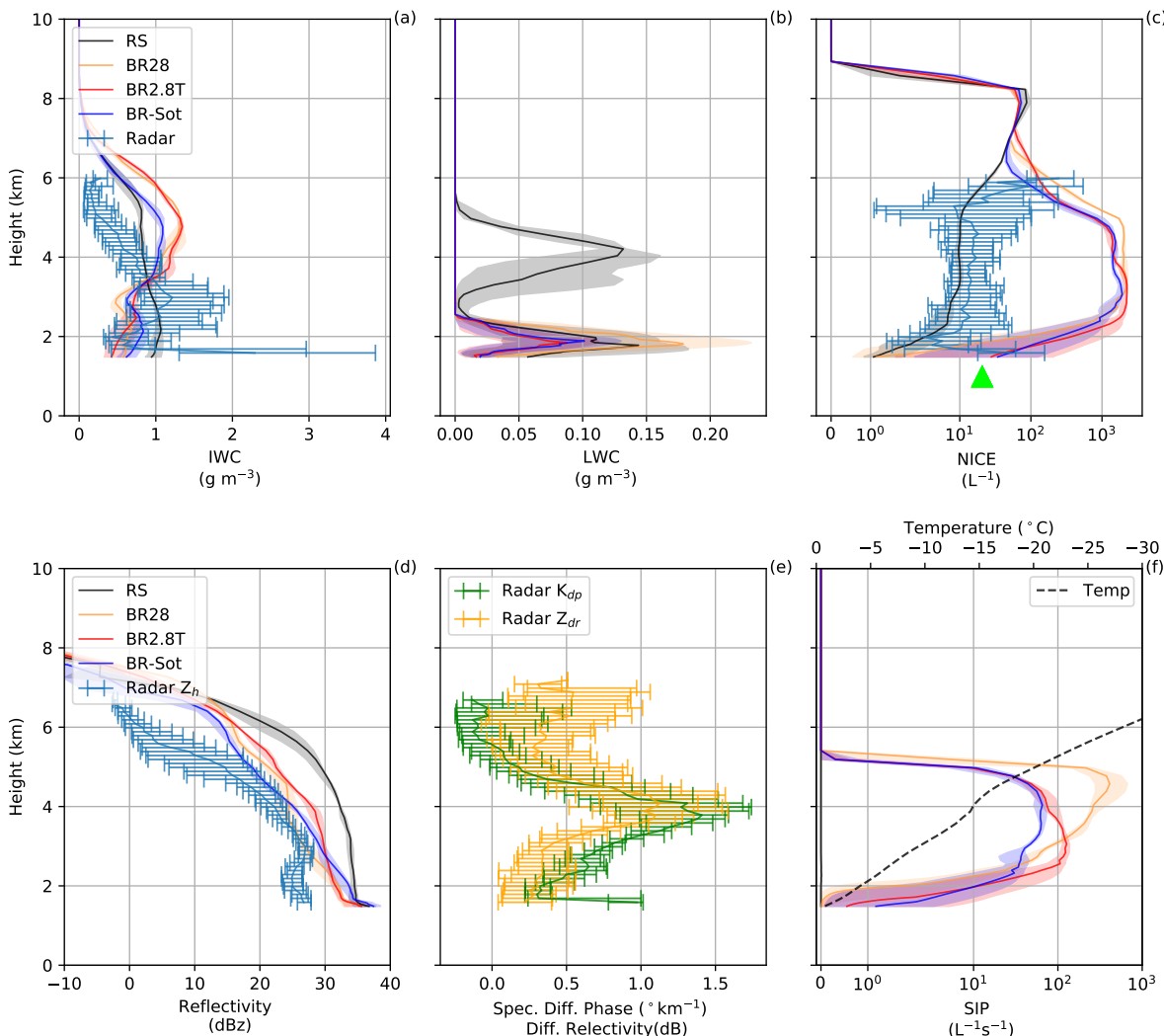

**Figure 7. (a)** Ice water content (IWC), **(b)** liquid water content (LWC), **(c)** ice number concentration (NICE), **(d)** model and radar reflectivity, **(e)** Specific differential phase ($K_{\mathrm{dp}}$), differential reflectivity ($Z_{\mathrm{DR}}$), and f) secondary ice production (SIP). The solid lines are the mean with shaded areas and errorbars showing the 10th and 90th percentiles for the model simulations and Doppler radar respectively at 15:30 UTC. The green triangle is the 2DVD surface observations for hydrometeors $D > 0.2$ mm.

Fig. 8. Regions in which SIP did not occur (e.g. $T < -21\,^\circ$C) were masked out for this analysis. Because the BR2.8T and BR-Sot simulations showed similar results, only the probability density functions (PDFs) for the wind variables from the BR2.8T simulation were analyzed.

A strong shift between the early and middle period in the V-wind median and interquartile range occurred from 20.5 to $-0.7\,\mathrm{m\,s^{-1}}$ and 6.6 to $21.3\,\mathrm{m\,s^{-1}}$, respectively compared to the U-wind (Table 1). In fact, the U-wind had a small variabil-

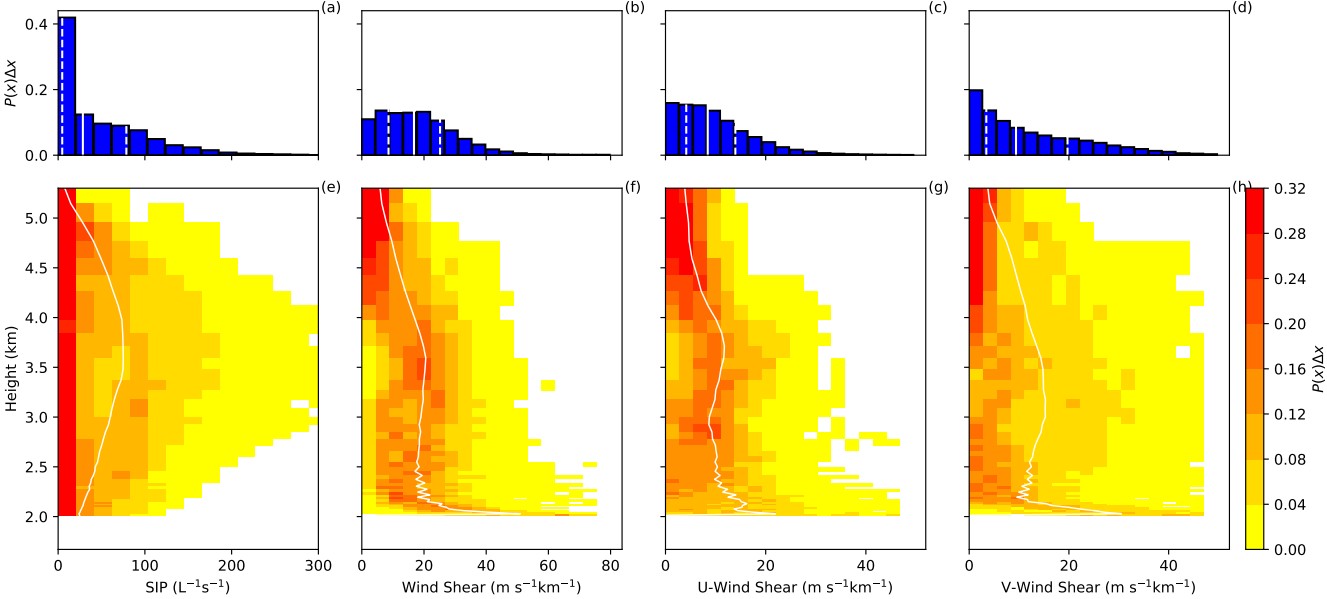

**Figure 8.** Probability density functions of different variables ($P(x)$) from the BR2.8T simulation from all model levels (top row) and at each model level (bottom row) for **(a, e)** Secondary Ice Production (SIP) rate, **(b, f)** Wind shear, **(c, g)** U-wind shear and **(d, h)** V-wind shear between 14:30-15:45 UTC. The solid and dashed white lines are the horizontal $50^{th}$ percentile and the $25^{th}$ and $75^{th}$ percentiles respectively of each variable over the $10\,\mathrm{km} \times 10\,\mathrm{km}$ domain.

ity between the early and middle periods (Table 1). As the afternoon progressed the median of the strongest V-wind shear extended from near the surface at 14:30 to 5 km amsl at 16:30 (Figs. 8 and 2). Updraft cells developed above this level of strong vertical wind shear between 10 and $20\,\mathrm{m\,s^{-1}\,km^{-1}}$. Our observation is consistent with Houze and Medina (2005) and Medina and Houze (2015) who also showed that updraft cells occurred at times and locations where the shear was strongest ($> \sim 10\,\mathrm{m\,s^{-1}\,km^{-1}}$). During the middle time period, the variability in the V-wind shear was the largest with an interquartile range of $16.3\,\mathrm{m\,s^{-1}\,km^{-1}}$ which coincided with the highest SIP rate (Fig. 8d and Table 1). The joint PDF ($P(\mathrm{SIP\ rate, V\text{-}wind\ shear})$) illustrates that the median of correlation between the V-wind shear and SIP rate peaked at $9\,\mathrm{m\ s^{-1}\,km^{-1}}$ and $80\,\mathrm{L^{-1}s^{-1}}$ (Fig. 10e). This peak coincided with the region where the wind shifted from south-westerly to northerly (along the valley) between the early and middle period. This shift was a result of the change in the V-wind speed from negative to positive at 2.9 km amsl. The joint PDF between the V-wind and SIP (Fig. 10 b) was highest at this altitude. Here, the strong and variable V-wind shear co-varied the best with the strong overall wind shear. The V-wind shear was the most important determinant of SIP of all the variables (Fig. 10f and Table 1).

On the other hand, the contribution of the updraft velocity to SIP is not as clear. The increase (early to middle period) and decrease (middle to late period) of the median of the SIP rates poorly co-varied with the U-wind shear or updraft compared to the V-wind shear (Figure 9). It appears that the V-wind shear played the major role in the SIP.

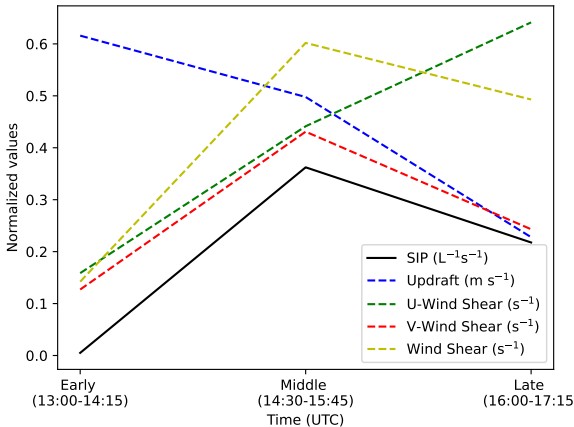

**Figure 9.** Normalized median values of the PDFs of the variables in Table 1 for the three different time periods for the BR2.8T simulation.

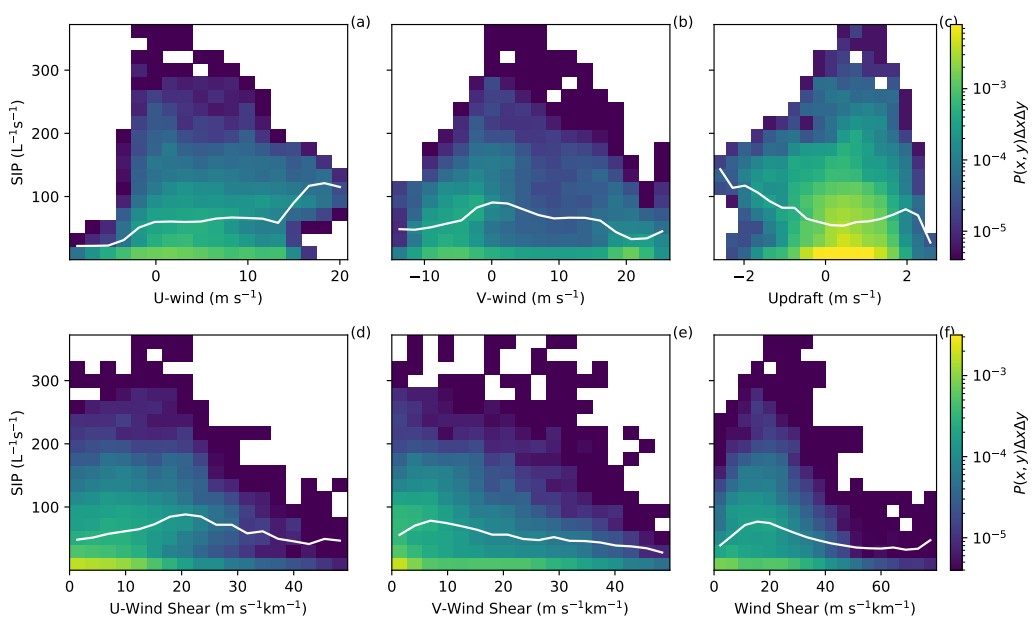

**Figure 10.** Joint probability density function multiplied by bin area ($P(x,y)\Delta x\Delta y$) for the model output of SIP versus **(a)** U-wind, **(b)** V-wind, **(c)** Updraft, **(d)** U-wind shear, **(e)** V-wind shear and **(f)** Wind shear for the BR2.8T simulation . White lines are the $50^{th}$ percentile as a function of the $x$-axis variable.

To further highlight this role, we calculated the mutual information shared between different sets of variables. The mutual information (MI: $I(X;Y)$) between variables $X$ and $Y$ was further analyzed for non-linear relationships (Shannon and Weaver, 1949) where $X \in$ [SIP rate] and $Y \in$ [U-wind, V-wind, Updraft, U-wind shear, V-wind shear, Wind Shear]. $I(X;Y))$ of 0 bits

**Table 1.** The 25th, 50th and 75th percentiles and the interquartile range (IQR) between the 10th and 90th percentiles of the vertical profiles for the BR2.8T simulations.

| Time (UTC) | Variable | 25th perc. | 50th perc. | 75th perc. | IQR. |
|---|---|---|---|---|---|
| 13:00-14:15 | SIP rate ($L^{-1}s^{-1}$) | 0.0 | 0.4 | 3.5 | 3.4 |
| | U-Wind ($m\,s^{-1}$) | 3.9 | 7.9 | 11.3 | 7.4 |
| | V-Wind ($m\,s^{-1}$) | 15.9 | 20.5 | 22.5 | 6.6 |
| | Wind Speed ($m\,s^{-1}$) | 16.3 | 22.9 | 24.8 | 8.5 |
| | Updraft ($m\,s^{-1}$) | 0.2 | 0.6 | 1.0 | 0.8 |
| | U-Wind Shear ($m\,s^{-1}\,km^{-1}$) | 2.0 | 4.3 | 7.4 | 5.5 |
| | V-Wind Shear ($m\,s^{-1}\,km^{-1}$) | 1.6 | 3.9 | 10.7 | 9.0 |
| | Wind Shear ($m\,s^{-1}km^{-1}$) | 4.1 | 7.1 | 14.8 | 10.7 |
| 14:30-15:45 | SIP rate ($L^{-1}s^{-1}$) | 4.7 | 28.5 | 78.7 | 73.9 |
| | U-Wind ($m\,s^{-1}$) | 0.8 | 4.2 | 8.6 | 7.8 |
| | V-Wind ($m\,s^{-1}$) | -5.5 | -0.7 | 15.8 | 21.3 |
| | Wind Speed ($m\,s^{-1}$) | 6.5 | 10.3 | 17.3 | 10.8 |
| | Updraft ($m\,s^{-1}$) | 0.1 | 0.5 | 0.9 | 0.8 |
| | U-Wind Shear ($m\,s^{-1}\,km^{-1}$) | 4.1 | 8.4 | 13.9 | 9.8 |
| | V-Wind Shear ($m\,s^{-1}\,km^{-1}$) | 3.4 | 9.4 | 19.7 | 16.3 |
| | Wind Shear ($m\,s^{-1}km^{-1}$) | 8.6 | 16.8 | 25.2 | 16.6 |
| 16:00-17:15 | SIP rate ($L^{-1}s^{-1}$) | 0.7 | 17.2 | 53.2 | 52.5 |
| | U-Wind ($m\,s^{-1}$) | 5.3 | 10.4 | 15.4 | 10.2 |
| | V-Wind ($m\,s^{-1}$) | -2.2 | -0.2 | 3.7 | 6.0 |
| | Wind Speed ($m\,s^{-1}$) | 7.4 | 12.8 | 16.6 | 9.2 |
| | Updraft ($m\,s^{-1}$) | -0.0 | 0.2 | 0.5 | 0.5 |
| | U-Wind Shear ($m\,s^{-1}\,km^{-1}$) | 5.9 | 11.3 | 16.5 | 10.6 |
| | V-Wind Shear ($m\,s^{-1}\,km^{-1}$) | 3.1 | 6.1 | 9.8 | 6.8 |
| | Wind Shear ($m\,s^{-1}km^{-1}$) | 10.3 | 14.5 | 19.6 | 9.3 |

means no information is shared between $X$ and $Y$ and therefore $Y$ cannot be inferred from $X$ (Further information about MI can be found in Appendix A and also in Dawe and Austin (2013).). For instance, during the last period the relationship of $I$(SIP rate; Wind shear) weakened drastically to 0.021 bits to below the level of significance (Table 2) compared to earlier periods. This was expected because the diminishing cloud liquid water caused a reduction in the riming rates. Lower riming rates limited graupel formation which in turn reduced ice-graupel collisions.

The significantly higher MI score between $I$(SIP rate; Wind shear) during the early and middle period, was a result of the strengthening of the northerly valley winds during the early afternoon hours when the predominant wind aloft was south-westerly (generating the dominant V-wind shear). The development of the northerly winds could have been a result low-level

**Table 2.** Mutual information between SIP rate and wind properties of the vertical profiles for the BR2.8T simulations. The significance level is calculated by taking the maximum of 1000 Monte Carlo simulations of mutual information between a random permutation of SIP rates and each variable.

| Time (UTC) | Variable | MI | Sig. level |
|---|---|---|---|
| 13:00-14:15 | $I$(SIP rate;U-Wind) | 0.025 | 0.009 |
| | $I$(SIP rate;V-Wind) | 0.037 | 0.009 |
| | $I$(SIP rate;Wind speed) | 0.029 | 0.009 |
| | $I$(SIP rate;Updraft) | 0.027 | 0.009 |
| | $I$(SIP rate;U-Wind Shear) | 0.008 | 0.011 |
| | $I$(SIP rate;V-Wind Shear) | 0.015 | 0.011 |
| | $I$(SIP rate;Wind Shear) | 0.011 | 0.009 |
| 14:30-15:45 | $I$(SIP rate;U-Wind) | 0.091 | 0.018 |
| | $I$(SIP rate;V-Wind) | 0.116 | 0.018 |
| | $I$(SIP rate;Wind speed) | 0.112 | 0.018 |
| | $I$(SIP rate;Updraft) | 0.035 | 0.018 |
| | $I$(SIP rate;U-Wind Shear) | 0.039 | 0.022 |
| | $I$(SIP rate;V-Wind Shear) | 0.043 | 0.021 |
| | $I$(SIP rate;Wind Shear) | 0.048 | 0.015 |
| 16:00-17:15 | $I$(SIP rate;U-Wind) | 0.105 | 0.054 |
| | $I$(SIP rate;V-Wind) | 0.117 | 0.054 |
| | $I$(SIP rate;Wind speed) | 0.103 | 0.054 |
| | $I$(SIP rate;Updraft) | 0.095 | 0.054 |
| | $I$(SIP rate;U-Wind Shear) | 0.014 | 0.067 |
| | $I$(SIP rate;V-Wind Shear) | 0.029 | 0.067 |
| | $I$(SIP rate;Wind Shear) | 0.021 | 0.055 |

blocking that occurred generating the shear layer (Medina et al., 2005). The sharp change in the wind speed and direction enhanced the turbulent overturning and therefore promoted the riming of ice crystals and snow leading to the formation of graupel which in turn enhanced the SIP rates. We hypothesise that if ice-ice collisions were included in the model configuration, the simulations could well have captured the intense $K_{dp}$ values in excess of $1.4\,^{\circ}\,\mathrm{km}^{-1}$ seen in Doppler observations after 365 16:40 UTC (Figs. 3a and S5e). Such a configuration may have even shown a better correlation between the vertical wind shear and ice-ice collisions from 17:00 UTC on-wards. The diminishing graupel number concentration limited the duration of SIP in our results and revealed a shortcoming in describing ice number concentrations through ice-graupel collisions when clouds enter a glaciated state.

### 3.4 SIP sensitivity to conversion rates

In this section the sensitivity of SIP to the rate of graupel formation, which is dependent on ice or snow crystals being larger than a given size when riming occurs, is analyzed. Fig. 11 shows the PSD for the sensitivity studies during which the size restrictions are modified which could slow the conversion process of the ice crystals and snow particles to graupel. The PSD over the cross-section at 15:30 UTC showed little difference in the ice crystal number concentrations where we expected higher ice crystal number concentration for BR2.8T and consequently higher snow number concentrations due to enhanced aggregation (Fig. 11a and b). The largest differences from the BR2.8T_300 simulation were in the form of enhanced snow (for diameters: $0.14 < \bar{D}_s < 0.42$ mm) and graupel number concentrations (for diameters: $1.2 < \bar{D}_g < 2.2$ mm). However, at 15:30 UTC there is no clear signal beyond model variability showing that the slower conversion rates to graupel affect the simulations (Fig. 11b and c). We compared the probability distributions of the total number of ice hydrometeors (NISG), SIP rate, ice crystal and graupel number concentrations of the BR2.8T_500, BR2.8T_400 and BR2.8T_300 simulations, respectively, to that of the reference simulation, BR2.8T between 14:15 to 15:45 UTC (Fig. 5b). We chose this time period because the largest graupel concentrations were present. The Kullback-Leibler divergence ($D_{KL}(P \| Q)$), which measures how one probability distribution $P$ is different from a second probability distribution, $Q$, shows little information loss between variables in Table 3, except for graupel. A value of 0 bits means that the probability distributions are the same (e.g., no information loss). The largest $D_{KL}(\text{BR2.8T\_300} \| \text{BR2.8T})$ was 3.33 bits for the graupel distribution and was reflected in larger differences in the SIP rate of of 1.07 bits and ice number concentration of 0.05 bits (Table 3 and Fig. 11c). If the SIP rate was sensitive to the conversion rate it is expected that the information loss would be the greatest between BR2.8T_500 and BR2.8T and not between BR2.8T_300 and BR2.8T (e.g., $D_{KL}(\text{BR2.8T\_500} \| \text{BR2.8T}) > D_{KL}(\text{BR2.8T\_400} \| \text{BR2.8T}) > D_{KL}(\text{BR2.8T\_300} \| \text{BR2.8T})$). This result leads us to conclude that the different conversion rates from ice crystals and snow to graupel, used in the paper in conjunction with the collisional breakup parameterization, were not significant for SIP.

## 4 Conclusions

A cold front passage on 26 March 2010, over the Swiss Alps, associated with strong vertical wind shear and intense polarimetric signatures was observed with a dual-polarization Doppler weather radar deployed at Davos. This study investigates the role of vertical wind shear on the rate of SIP by making simulations of wintertime orographic MPCs with a non-hydrostatic, limited area model, COSMO, which has a two-moment bulk microphysics scheme with six hydrometeor categories, and two additional parameterizations for ice-graupel collisions (e.g., Sotiropoulou et al., 2020; Dedekind, 2021) based on Takahashi et al. (1995). To conclude, our main finding can be summarized as follows:

- Large values of $K_{dp} > 1\,^\circ\,\text{km}^{-1}$ suggest that a large population of oblate particles was present, most likely originating from ice-ice collisions, at 4 km amsl. This level coincided with the $-15^\circ$C isotherm. At $-15^\circ$C dendritic growth is very fast causing low-density dendrites to fracture and aggregate. At this time, also $Z_{DR}$ was positive, indicating that

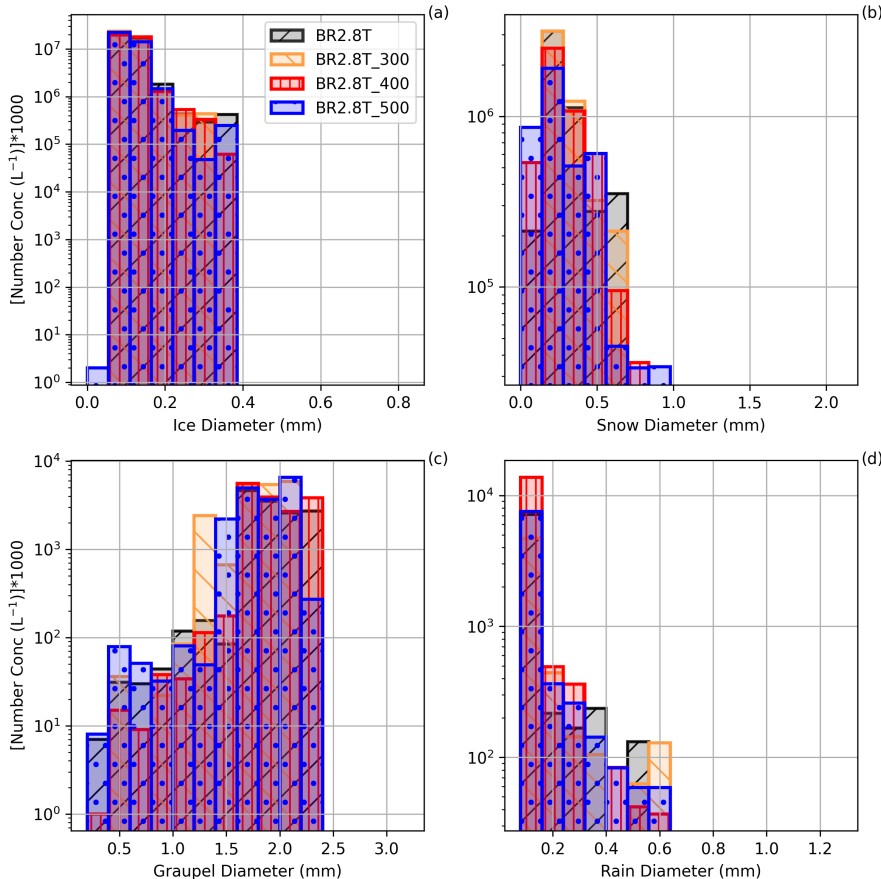

**Figure 11.** Particle size distribution over the number concentrations for panels **(a)** ice, **(b)** snow, **(c)** graupel, **(d)** cloud droplets and **(e)** raindrops for all the sensitivity simulations at 15:30 UTC. BR2.8T, BR2.8T_300, BR2.8T_400 and BR2.8T_500 represents the size restriction of 200, 300, 400 and 500 μm respectively before ice crystals and snow can be converted to graupel.

large isotropic particles were less present. At lower altitudes $Z_H$ increased while $K_{dp}$ (and $Z_{DR}$) decreased suggesting aggregation and/or riming were occurring.

– The rime splintering simulations overestimated the $Z_H$ throughout the vertical profile and underestimated the disdrometers' number concentration of hydrometeors at the surface. Both shortcomings could be explained by omission of ice-graupel collisions.

– The breakup simulations (BR28, BR2.8T and BR-Sot) caused narrower ice crystal and snow distributions (enhanced number concentrations of smaller ice particles) resulting in a better representation of $Z_H$. The enhanced number concentrations of ice particles meant that these simulations, in particular BR2.8T and BR-Sot, captured the disdrometer observations of $\sim 20\,\mathrm{L}^{-1}$ (considering the 0.2 mm observation limit) at 15:30 and 17:00 UTC.

**Table 3.** The Kullback-Leibler divergence, $D_{KL}(P \| Q)$, between two probability distributions $P$ and $Q$ from 14:14 to 15:45 UTC. $P$ (BR2.8T_300, BR2.8T_400 and BR2.8T_500) is the measured probability distribution against the reference probability distribution $Q$ (BR2.8T). Each distribution consist of $\sim$10800 grid points separated into 24 bins over the cross-section in Fig. 1.

| Variable | $P$ | $Q$ | $D_{KL}(P \| Q)$ (bits) |
|---|---|---|---|
| NISG | BR2.8T_300 | BR2.8T | 0.06 |
| | BR2.8T_400 | | 0.03 |
| | BR2.8T_500 | | 0.03 |
| SIP | BR2.8T_300 | BR2.8T | 1.07 |
| | BR2.8T_400 | | 0.06 |
| | BR2.8T_500 | | 0.06 |
| Ice | BR2.8T_300 | BR2.8T | 0.05 |
| | BR2.8T_400 | | 0.02 |
| | BR2.8T_500 | | 0.03 |
| Graupel | BR2.8T_300 | BR2.8T | 3.33 |
| | BR2.8T_400 | | 0.84 |
| | BR2.8T_500 | | 0.61 |

– During the middle period, 14:30-15:45 UTC, the V-wind shear increased by 60% causing conditions favorable for accretion leading to enhanced graupel formation and SIP in the region favorable for SIP.

– Another time period with high $K_{dp}$, but low $Z_{DR}$ was observed at 17:00 which was not captured by the breakup simulations because the graupel mixing ratio was depleted. The breakup parameterization does not include ice-ice collisions and relies only on graupel as the collider specie. At this time the radar signatures suggested that dendrite collisions caused the formation of small oblate particles (increasing $K_{dp}$) but also the formation of a few, larger, isotropic aggregates (decreasing $Z_{DR}$).

– The mutual information between the SIP rate and other variables like vertical wind shear and updraft velocity suggested that the SIP rate is best predicted by the overall wind shear.

– The sensitivity of the ice-graupel simulations to the conversion rate size restriction was measured using the Kullback-Leibler divergence. Ice and snow (with diameters of 300 μm) that were converted to graupel showed the biggest deviation from the reference size of 200 μm. However, the sensitivity simulations were not sensitive to the conversion rate size restriction.

Turbulent overturning, whether it is associated with baroclinic waves (Gehring et al., 2020) or low-level blocking (Medina et al., 2005; Houze and Medina, 2005; Medina and Houze, 2015), has been shown to play an important role in accreting hydrometeors to form precipitation. Here, we considered that the interactions of ice hydrometeors can lead to ice-graupel

collisions, causing enhanced small ice fragments, as opposed to only growing larger through aggregation. These smaller fragments fall slower against updraft and may decrease local precipitation rates enhancing precipitation downstream of the flow (Dedekind et al., 2021). Wind shear plays a significant role in ice-graupel collisions and may even be more important when all ice-ice collisions are considered in more physically robust collisional breakup parameterizations (Yano and Phillips, 2011; Phillips et al., 2017). By only considering ice-graupel collisions we are limited to mainly investigating collisional breakup in

MPCs where riming can occur to form graupel. In the case where a cloud becomes glaciated and graupel cannot form through riming, our parameterization will not be able to simulate SIP, which may still prove to be very important.

## 5 Data availability

The COSMO model output, radar and 2DVD datasets used for our analysis are available at https://doi.org/10.5281/zenodo.6609251 and the software to analyze the data can be found at https://doi.org/10.5281/zenodo.6612296

## Appendix A: Derivation of the simulated radar reflectivity

Here we briefly show the calculation of the simulated $Z_{\mathrm{H}}$ (Appendix B, Seifert, 2002). Calculating $Z_{\mathrm{H}}$ from the two-moment cloud microphysics scheme would not be possible without approximations and assumptions. The following relationship for the radar reflectivity of drops ($Z_w$), using the Rayleigh approximation for the cross-section of drops (Eq. A1), results in:

$$\eta_w = \frac{\pi^5 |K_w|^2}{\lambda_R^4} \int_0^\infty D^6 f_w(D) dD \tag{A1}$$

where $D$ is the particle diameter, $\lambda_R$ is the wavelength of radar radiation, $\eta_w$ is the volumetric liquid water content, $f_w(D)$ is the number density distribution function for liquid water and $K_w^2 = 0.93$ the dielectric constant of liquid water. The reflectivity factor for cloud water is given by:

$$\tilde{Z}_w = \frac{\lambda_R^4}{\pi^5 |K_w|^2} \eta_w = \int_0^\infty D^6 f_w(D) dD = \left(\frac{6}{\pi \rho_w}\right)^2 \int_0^\infty x^2 f_w(x) dx = \left(\frac{6}{\pi \rho_w}\right)^2 Z_w \tag{A2}$$

where $\rho_w$ is the water density. Because of the backscatter behavior for the mass-equivalent diameter with regards to $\rho_w$ and $\rho_i$

(ice density), the same applies to graupel, which is described as a spherical ice particle

$$\eta_g = \frac{\pi^5 |K_i|^2}{\lambda_R^4} \left(\frac{6}{\pi \rho_i}\right)^2 \int_0^\infty x^2 f_g(x) dx_1 \tag{A3}$$

where $x$ is the particle mass, $f_g(x)$ is the number density distribution function for graupel and $K_i^2 = 0.176$ is the dielectric constant of ice. The radar reflectivity factor for ice particles (e.g., graupel) is given by:

$$\tilde{Z}_g = \frac{\lambda_R^4}{\pi^5 |K_w|^2} \eta_g = \frac{|K_i|^2}{|K_w|^2} \left(\frac{6}{\pi \rho_i}\right)^2 \int_0^\infty x^2 f_g(x) dx = \frac{|K_i|^2}{|K_w|^2} \left(\frac{6}{\pi \rho_i}\right)^2 Z_g. \tag{A4}$$

For melting ice particles, however, $K_w$ must be used instead of $K_i$. In our study, the surface and in-situ cloud temperatures were below $0\,^\circ$C. Therefore, more information on the reflectivity calculations for melting ice particles can be found in Seifert (Appendix B, 2002). Finally, the radar reflectivity factor is given by:

$$dBZ = \frac{10}{\ln 10} \ln\left[\frac{Z_{radar}}{\mathrm{mm^6\ m^{-3}}}\right] \tag{A5}$$

where $Z_{radar}$ is the sum of the reflectivity calculated for each individual cloud particle category (e.g., cloud drops, raindrops,
ice crystals, snow crystals, graupel and hail):

$$Z_{radar} = \left(\frac{6}{\pi\rho_w}\right)^2 \left[Z_c + Z_r + \frac{\rho_w^2}{\rho_i^2}\frac{|K_i^2|}{|K_w{}^2|}\left(Z_{ic} + Z_s + Z_g\right)\right]. \tag{A6}$$

## Appendix B: Mutual Information

The entropy $H$ of the variable $x$'s probability density function $P(x)$ is defined by Shannon and Weaver (1949) to be:

$$H = -\int P(x)\ln(P(x))\mathrm{d}x \tag{B1}$$

where $x$ is the information content of a single measurement of $P(x) = -\ln P(x)$. The entropy is a measure of the amount of information that is required to represent the PDF. From here both the Kullback-Leibler divergence and the mutual information can be claculated.

The Kullback-Leibler divergence, also known as the relative entropy, measures the distance between two probability distributions, $P(x)$ and $Q(x)$ over a discrete random variable $X$. The Kullback-Leibler divergence is defined as follows:

$$D_{\mathrm{KL}}(P\,\|\,Q) = \int P(x)\ln\left(\frac{P(x)}{Q(x)}\right)\mathrm{d}x. \tag{B2}$$

.

The mutual information (MI) is a measure of the mutual dependence between two random variable $X$ and $Y$ (e.g., the entropy of $X$ subtracted from the entropy of $X$ conditioned on $Y$):

$$I(X;Y) = H(X) - H(X|Y). \tag{B3}$$

MI describes therefore how different the joint distribution of the pair $(X,Y)$ is from the distribution of $X$ and $Y$. Combining equations (B1) and (B3) yield:

$$I(X;Y) = \int [P(x)\ln(P(x)) - P(x,y)\ln(P(x|y))]\mathrm{d}x\mathrm{d}y \tag{B4}$$

.

and because $P(x|y) = P(x,y)/P(y)$, equation (B5) can be reduced to:

$$I(X;Y) = -\int P(x,y) \ln\left(\frac{P(x,y)}{P(x)P(y)}\right) \mathrm{d}x\mathrm{d}y. \tag{B5}$$

The range of the MI is described as follows:

$$\mathrm{MI} = \begin{cases} 0, \text{ if } P(x,y) = P(x)P(y), & (X \text{ and } Y \text{ are completely independent}) \\ H(X), \text{ if } P(x,y) = P(x) = P(y), & (X \text{ and } Y \text{ are perfectly correlated}) \end{cases} \tag{B6}$$

*Author contributions.* ZD conducted the simulations and JG post-processed on the radar data. ZD and JG were main authors of the paper and interpreted the data. PA and UL contributed to the interpretation and analysis techniques. ZD and JG contributed to the writing of the study.

*Competing interests.* The authors declare that they have no conflict of interest.

*Acknowledgements.* We would like to thank Alexis Berne for his valuable feedback on the interpretation of the radar data. All simulations were performed with the Consortium for Small-scale Modeling (COSMO) model. The simulations were performed and are stored at the Swiss National Supercomputing Center (CSCS) under project s1009. ZD and JG acknowledge funding from the Swiss National Science Foundation (SNSF), grant numbers 200021_175824 and 200020_175700, respectively. We sincerely thank the reviewers for their constructive feedback. Their suggestions and comments considerably improved the quality of the manuscript.

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
