# Peer review of "Heavy snowfall event over the Swiss Alps: Did wind shear impact secondary ice production?"

_Atmospheric Chemistry and Physics, 2022_

## Referee Comment (RC1)

**Heavy snowfall event over the Swiss Alps: Did wind shear impact secondary ice production?**

General comments

This article puts forward the interesting idea that wind shear could "enhance interaction between ice particles", particularly through shear-induced turbulence, and promote secondary ice production. There are other interesting ideas also, for example that secondary ice production strength or dependencies could be hidden behind unrealistic hydrometeor growth rates. It is challenging and important to compare high-resolution model output and ground-based observations of microphysical variables, and I appreciate that effort here.

Despite these positive aspects, there is significant work to be done on this manuscript before it is in publishable form. The discussion of results was quite difficult to follow, in part because the figures do not seem to be ordered logically and in part because the writing is often convoluted. Points 3 and 4 below are two suggestions to improve this in my opinion. Before the results are restructured a bit, it is hard for me to know if I am convinced by them.

1. I felt that a small, general overview of the use of remote sensing to study secondary ice production in general, or ice-ice collisional breakup in particular, was missing from the Introduction. For example, I think the recent study of Luke et al. 2021 using longer-term ground-based remote sensing to infer secondary ice production strength would be worth mentioning.

2. The two prior studies mentioned (Grazioli et al. 2015a and von Terzi et al. 2022) are also employing specific differential phase shift, which is not defined in terms of its information content. Only differential and horizontal reflectivity are defined. I think the definitions in Section 4 of Field et al. 2017 are quite nice; perhaps some variant of those could be included here. ("By transmitting horizontally and vertically polarized waves and looking at the differences in power and phase between the echoes in each polarization, information about the orientation and/or phase of the hydrometeors being probed can be obtained." "Just as the backscatter is different for horizontal and vertical polarizations in the presence of oriented ice crystals, so too is the speed at which the radar wave propagates through the cloud. This leads to a small phase shift between the horizontal and vertical polarized echoes.")

3. I had some difficulty to follow the arguments in Section 3.1.1. I am thinking that the RS process is muted in the simulation because there are insufficient droplets of the correct size. Then depositional growth or aggregation dominates growth and broadens the ice crystal size distributions, promoting the size sorting? But a droplet limitation is not explicitly mentioned, and to me the chain of events in the model is "limited riming + strong aggregation / depositional growth → broad ice size distributions → size sorting", whereas the size sorting is described first in the text before its causes. Could the ideas be rearranged in this subsection to follow the argument?
    a. As a sub-comment here, I am still confused by the contradicting reflectivity (Fig 2) and ice crystal number (Fig 4) results. How can the reflectivity from the RS simulation be so off when its crystal number and IWC are reasonably accurate? This is really just the product of a "shortcoming in the derivation of NICE from the radar obs"?

4. Again, in the ordering of results, it would have made more sense to me to show Figure 7 and some of the results in Section 3.2 prior to any cloud fields. It is a bit hard to tell from the colorbar in Figure 7, but it seems to me that midlevel (~3-5 km) wind speeds are being overestimated pretty much by the model. If there are strong biases in the wind field, then we cannot expect agreement in the cloud (microphysical) fields.

Specific comments

**Line 78-79** Again, a definition of $K_{dp}$ and a basis for comparison for values of 1.5° and 2° km$^{-1}$ would be helpful for readers who are non-experts in radar.

**Line 98-99** Would it be too cumbersome to include the formulations (in some condensed form) of Murphy et al. (2020) to convert $Z_H$, $Z_{DR}$, and $K_{DP}$ to microphysical quantities here? It would give the reader a better idea of how the measurements are being used.

**Line 118** It seems to me that the motivation for including only ice-graupel collisions (not, for example, snow-ice collisions) is the theoretical constraint from Phillips et al. 2017 for a sufficient collision kinetic energy. Perhaps this should be explicitly mentioned.

**Lines 139** Is the temperature threshold correct here? Normally, a threshold freezing of cloud droplets occurs at -37°C not -50°C.

**Lines 144-145** Confusing wording here. How about "As for many other numerical weather models, rime splintering is the only secondary ice production process included in the standard version of COSMO." Also, I assume the RS formulation uses the general 350 fragments per milligram rime value, but I would mention this value here.

**Lines 169-170** Could you say something more precise about what you mean by *early graupel formation*? As it stands, you say that it is "promoted when ice crystals or snow are converted to graupel" which is a bit of a tautology.

**Lines 190-191** "The vertical evolution of $K_{dp}$ and $Z_{DR}$ is similar, with a peak observed about 4 km amsl.." Here you are already looking at Figure 5, right? Please cite the figure.

**Line 208-209** Unless I am missing something, I would remove the sentence that IWC and NICE "fall within the 10 and 90$^{th}$ percentiles range of the observations." This does not really indicate any agreement to me.

**Line 248-250** Given that no simulation performs best on all metrics, is it a fair conclusion that the size scaling from Sotiropoulou et al. 2021b (your Equation 3) is not an important factor in this case?

**Figure 6** Which simulation is this figure from?

**Line 283** I would remove the "not surprisingly" here. There has been significant discussion of how updraft modulates SIP rates but not shear, so it is indeed surprising that longitudinal wind shear is the "most important determinant" here.

**Line 349** I would write "Both shortcomings *could* be explained by omission of ice-graupel collisions." There are also other processes that could explain an overestimation of $Z_H$ and underestimation of NICE.

**Line 296-297** "The higher MI values for V-wind shear with SIP is most likely why the Wind shear had larger and significant MI values with SIP." I do not understand this; to me, it sounds like you are saying the values are larger because they are larger. Could you please reword or remove?

Small editorial stuff
**Line 53** prevelent → prevalent

**Line 79** e.g. removed. 252 K = -21° C; it's not an example.

**Line 82** number high → high number. Also, "as *large* $K_{DP}$ is an indicator of.." Only large values of $K_{dp}$ indicate high number concentrations of oblate hydrometeors.

**Line 154** recoreded → recorded

**Figures 2/3/7 caption** Hofmoller → Hovmöller*

**Line 359** specie → species

References

P. Field et al. (2017) Secondary ice production: Current state of the Science and Recommendations for the Future. *Meteor. Monog.* **58** 10.1175/AMSMONOGRAPHS-D-16-0014.1

E. P. Luke et al. (2021) New insights into ice multiplication using remote-sensing observations of slightly supercooled mixed-phase clouds in the Arctic. *Proc. Nat. Acad. Sci.* **118** (13) e2021387118.

V. T. J. Phillips et al. (2017) Ice multiplication by breakup in ice-ice collisions. Part I: Theoretical Formulation. *J. Atm. Sci.* **74** (6) 10.1175/JAS-D-16-0224.1

---

## Author Comment (AC1)

**Heavy snowfall event over the Swiss Alps: Did wind shear impact secondary ice production?**

Zane Dedekind[*1], Jacopo Grazioli[*2], Philip H. Austin[1], and Ulrike Lohmann[3]

[1]Department of Earth, Ocean, and Atmospheric Sciences, University of British Columbia, Earth Sciences Building, 2207 Main Mall, Vancouver, BC, V6T 1Z4, Canada
[2]Environmental Remote Sensing Laboratory (LTE), École Polytechnique Fédérale de Lausanne (EPFL), Lausanne, Switzerland
[3]Institute of Atmospheric and Climate Science, ETH Zurich, Switzerland
[*]Equally contributing authors

We sincerely thank Reviewer 1 for the constructive feedback. The suggestions and comments considerably improved the quality of the manuscript.

Below we present a detailed response with the reviewer comments in black, our responses in blue and additions to the manuscript in blue italics.

**General comments**

1. I felt that a small, general overview of the use of remote sensing to study secondary ice production in general, or ice-ice collisional breakup in particular, was missing from the Introduction. For example, I think the recent study of Luke et al. (2021) using longer-term ground-based remote sensing to infer secondary ice production strength would be worth mentioning.

   We agree with the suggestion of the reviewer and with the fact that an overview of the use of remote sensing to study SIP is needed. We expanded the introduction to include an overview of the subject. We refer now to Zawadzki et al. (2001), Oue et al. (2015), and Luke et al. (2021) (highlighting however that their approach made use of Doppler spectra, not available to us, rather than dual-polarization bulk measurements. For this second (our) approach we refer to Hogan et al. (2002); Andrić et al. (2013); Sinclair et al. (2016); Kumjian and Lombardo (2017). This part of the Introduction in the revised manuscript now reads:

   *Remote sensing from weather radars has been used to study snowfall microphysics and hydrometeors' habit (e.g., shape, phase or hydrometeor type). Although radar observations do not provide a direct information on SIP, a few studies leveraged the Doppler and/or dual-polarization capabilities of weather radars to identify the occurrence of SIP and to speculate case-by-case on the possible mechanisms behind its origin. Two non mutually exclusive approaches can be found in the literature. Zawadzki et al. (2001); Oue et al. (2015); Luke et al. (2021) exploited Doppler spectra collected by vertically-pointing radars to identify the appearance of secondary populations of particles at given altitudes or tem-*

*perature levels. Other approaches (Hogan et al., 2002; Andrić et al., 2013; Sinclair et al., 2016; Kumjian and Lombardo, 2017) focused on the interpretation of the signature of dual-polarization variables and their respective evolution over the vertical column of precipitation. This second approach, used in this study, leverages the fact that dual-polarization variables are complementary and affected in a different way by changes in number, shape, size and density of hydrometeors.*

2. The two prior studies mentioned (Grazioli et al., 2015b; von Terzi et al., 2022) are also employing specific differential phase shift, which is not defined in terms of its information content. Only differential and horizontal reflectivity are defined. I think the definitions in Section 4 of Field et al. (2016) are quite nice; perhaps some variant of those could be included here. ("By transmitting horizontally and vertically polarized waves and looking at the differences in power and phase between the echoes in each polarization, information about the orientation and/or phase of the hydrometeors being probed can be obtained." "Just as the backscatter is different for horizontal and vertical polarizations in the presence of oriented ice crystals, so too is the speed at which the radar wave propagates through the cloud. This leads to a small phase shift between the horizontal and vertical polarized echoes.")

The reviewer has a valid point here. $K_{dp}$ was not properly defined and the overall introduction of the polarimetric variables was not adapted to a broad audience (as we would like this manuscript to be understandable both for experts of the numerical model and remote sensing communities). This part of the Introduction section is rewritten in these terms:

*Not only the backscattered power is different for horizontal and vertical polarizations in the presence of anisotropic particles, but also the propagation speed of the waves. The rate of change in phase shift between the horizontal and vertical polarized echoes is expressed by the specific differential phase shift $K_{dp}$ [$°\mathrm{km}^{-1}$]. This variable is complementary and not redundant: it is in fact not affected by the absolute calibration of a radar and is less affected than $Z_{DR}$ by eventual presence of large isotropic particles within the sampling volume. Local $K_{dp}$ enhancements in snowfall have been documented (Schneebeli et al., 2013; Bechini et al., 2013) and in some cases associated to SIP (e.g., Andrić et al., 2013; Grazioli et al., 2015a; Sinclair et al., 2016).*

3. I had some difficulty to follow the arguments in Section 3.1.1. I am thinking that the RS process is muted in the simulation because there are insufficient droplets of the correct size. Then depositional growth or aggregation dominates growth and broadens the ice crystal size distributions, promoting the size sorting? But a droplet limitation is not explicitly mentioned, and to me the chain of events in the model is "limited riming + strong aggregation / depositional growth -> broad ice size distributions -> size sorting", whereas the size sorting is described first in the text before its causes. Could the ideas be rearranged in this subsection to follow the argument?

Yes, we agree with the chain of events you have pointed out and adapted the section with the following text:
*$Z_H$ was significantly overestimated by the RS simulation between 13:00 and 17:30 UTC which most likely was a result of the following chain of events. 1) Insufficient droplets of size 25 µm (Fig. 5d), within the narrow temperature range*

*($-3 \geq T \geq -8\,°C$), led to a limitation in ice particle growth by riming and therefore limited rime splintering (Fig. 6f). 2) Because rime splintering was not that active, typical for wintertime MPCs (e.g., Henneberg et al., 2017; Dedekind et al., 2021), the ice and snow crystals grew mainly by depositional growth and aggregation. 3) The ice and snow crystal size distributions widened substantially (Figs. 6a, b and S2a, b). These categories both had number concentrations less than $100\,L^{-1}$ with particle diameters of up to 0.8 and 5.1 mm, respectively, at 15:30 UTC. 4) The larger ice and snow crystal diameters resulted in enhanced $Z_H$. These observation is consistent with other times during the day which showed even larger sized ice and snow crystals of 0.9 and 5.2 mm, respectively (Figs. S3a and S4a) as well as higher rain mass mixing ratios (e.g. Fig. 5a). There were single grid points where snow crystal even reached diameters of 13 to 17 mm during the latter part of the day (not shown here). Additionally, excessive size sorting in the model most likely contributed to the overestimation in $Z_H$. Size sorting typically occurs within the sedimentation parameterization of 2M schemes in regions of vertical wind shear or updraft cores (Milbrandt and McTaggart-Cowan, 2010; Kumjian and Ryzhkov, 2012). All these factors contributed to the RS simulation overestimating $Z_H$ by at least 8 dBz throughout the vertical profile compared to the observations (Fig. 7d).*

(a) As a sub-comment here, I am still confused by the contradicting reflectivity (Fig 2) and ice crystal number (Fig 4) results. How can the reflectivity from the RS simulation be so off when its crystal number and IWC are reasonably accurate? This is really just the product of a "shortcoming in the derivation of NICE from the radar obs"?

There are a several reasons why the reflectivity in the RS simulation is overestimated: 1) Rime splintering was very low, which is typical for wintertime MPCs (Fig. 7f here and e.g., Henneberg et al., 2017; Dedekind et al., 2021), the ice and snow crystals grew mainly by depositional growth and aggregation to much larger sizes compared to the collisional breakup simulations (Fig. 6c). The snow crystals, especially, were double the size (Fig. 6b, over 2 mm in diameter). Below we show Hovmöller plots (Figs. 1 and 2) of the diameters for ice crystals, snow, and graupel from 15:00 to 17:30 UTC showing the snow and graupel particles reaching diameters of over 4 mm and 8 mm, respectively. 2) The rain mass and number concentrations were also substantially higher in the RS simulation. All these factors may have contributed to the large overestimation of $Z_H$ apart from the issues that arise from size sorting which is discussed in the manuscript. The adapted manuscript now includes Figure 3 showing $K_{dp}$, $Z_{DR}$ and $Z_H$ over the entire period. The large $K_{dp}$ values (see the reference values reply on the Specific Comments section point 1) implies large number concentrations of ice particles which is not consistent with the RS simulation (e.g., at 15:30 UTC as shown in Fig. 6). The observed ice crystal number concentrations, collected at the surface by the disdrometer, are also not in agreement with the RS simulation. These reasons indicate that the reflectivity will be overestimated in the RS simulation. We now discuss to the main assumptions and limitations which can cause shortcomings in the derivation of the IWC and NICE from the radar observations in Section 2.1.

4. Again, in the ordering of results, it would have made more sense to me to show Figure 7 and some of the results in Section 3.2 prior to any cloud fields. It is a bit hard to tell from the colorbar in Figure 7, but it seems to me that midlevel

[Figure]

**Figure 1.** Hovmöller diagrams of snow diameters for panels **(a)** RS , **(b)** BR28, **(c)** BR2.8T and **(d)** BR-Sot between 15:00 and 17:30 UTC.

( 3-5 km) wind speeds are being overestimated pretty much by the model. If there are strong biases in the wind field, then we cannot expect agreement in the cloud (microphysical) fields.

We agree in presenting these results earlier in the manuscript. Concerning the wind biases, there certainly appear to be stronger biases in the wind fields and the reviewer is certainly correct in raising the issue that expecting agreement in the microphysical fields is futile if the wind fields do not compare well. However, the interpretation of the model data should be interpreted in the following light: 1) the model output is used to generate three cross-sections (the entire width of the three cross-sections is $\approx 3$ km wide) which are interpolated along the direction of the RHI of the Doppler radar (Fig. 1). The mean of the three cross-sections yields the mean cross-section for each simulation. Using this method may generate biases; 2) Comparing results close to the surface in valleys and on the peaks of mountains over complex terrain is challenging because of the differences in the model topography and the actual topography (e.g., Goger et al., 2016). E.g., the altitude of the Doppler radar location (9.843°E, 46.789°N) is 2132.5 m compared to the closest model grid point (9.837°E, 46.79°N) of 1729 m. Additionally, the Doppler velocity also includes the falling z-oriented component of the hydrometeors which can generate more uncertainty in the comparison to the simulations.

**Specific comments**

[Figure]

**Figure 2.** The same as Figure 1, but for graupel.

1. Line 78-79 Again, a definition of Kdp and a basis for comparison for values of $1.5°$ and $2°\,\mathrm{km}^{-1}$ would be helpful for readers who are non-experts in radar.

   We decided to provide some reference values to put these number into context. We now cite the work of Schneebeli et al. (2013), who compiled a statistical analysis of polarimetric variables for this exact location and this radar system for a time span which includes the current case study.

   *$K_{\mathrm{dp}}$ scales with radar frequency. A statistical analysis of $K_{\mathrm{dp}}$ in snowfall conducted with this radar and in this location over a long observation period (Schneebeli et al., 2013), showed that the 80% quantile of $K_{\mathrm{dp}}$ at every height level is lower than $0.5°\,\mathrm{km}^{-1}$. Considering that the distribution of $K_{\mathrm{dp}}$ is very skewed, values above $1°\,\mathrm{km}^{-1}$ in snow can be considered as unusually large.*

2. Line 98-99 Would it be too cumbersome to include the formulations (in some condensed form) of Murphy et al. (2020) to convert $Z_H$, $Z_{DR}$, and $K_{dp}$ to microphysical quantities here? It would give the reader a better idea of how the measurements are being used.

We agree with the reviewer and we decided to provide the explicit formulation of the variables as well as a more thorough discussion of the (significant) sources of uncertainty for these relations. Eqs. 1, 2 and 3 in the revised manuscript present the mathematical formulation, while the discussion on the underlying assumptions and limitations reads:

*More details about the derivation of the equation can be found in Ryzhkov and Zrnic (2019); Murphy et al. (2020) but it is worth to focus on the main assumptions and limitations. The main assumptions are:*

- *The equations are derived assuming to be in the Rayleigh regime (which may not be fulfilled) for the X-band radar for large hydrometeors.*
- *The density and the size of the hydrometeors are assumed to be inversely proportional.*

*The retrievals have shown to be most reliable at $T < -10\,°C$, for low riming degrees and in regions where the $K_{dp}$ and $Z_{DR}$ signals are not close to 0. As recognized by Murphy et al. (2020), the errors may be large and in situ validation efforts are needed to refine these techniques. As a final caveat, the equations developed on theoretical basis are in practice very sensitive to the accuracy of the polarimetric variables, which can be very noisy. $K_{dp}$ in particular is an estimated variable affected by mean errors on the order of 30% (Grazioli et al., 2014).*

3. Line 118 It seems to me that the motivation for including only ice-graupel collisions (not, for example, snow-ice collisions) is the theoretical constraint from Phillips et al. (2017) for a sufficient collision kinetic energy. Perhaps this should be explicitly mentioned.

   This is not the case. It is rather a theoretical constraint from Sullivan et al. (2018). The parameterization is based of the ice fractures that are generated during the collisions of large rimed ice particles (e.g., graupel or hail) in the study of Takahashi et al. (1995). We added the following text to make the limitation explicit.
   *Because the parameterization from Sullivan et al. (2018) is based on experimental results by Takahashi et al. (1995), it is constrained to only ice-graupel collisions and may not be adequate when studying ice-ice collisions in winter-time MPCs consisting of mainly ice and snow crystals. The amount of fractures that can be generated in snow-ice collisions might, on the contrary, not be significant because of the low collision kinetic energy between unrimed particles. Experimental studies are thus required to show evidence for generated ice fractures between unrimed ice particles.*

4. Lines 139 Is the temperature threshold correct here? Normally, a threshold freezing of cloud droplets occurs at -37°C not -50°C.
   Correct. We offer the following explanation to make our statement clear. The temperature threshold at which most cloud droplets freeze is $\approx -38\,°C$. In COSMO, there is an additional step stating that all cloud drops should freeze at $\approx -50\,°C$. We adapted the manuscript:
   *Homogeneous freezing of cloud droplets, parameterized from the homogeneous freezing rates of Cotton and Field (2002), is calculated for $0 > T \geq -50\,°C$. At $-38\,°C$ most cloud droplets will freeze given the enhanced homogeneous nucleation rates at colder temperatures. As a lower bound the homogeneous freezing of all cloud droplets occurs at $T = -50\,°C$.*

5. Lines 144-145 Confusing wording here. How about "As for many other numerical weather models, rime splintering is the only secondary ice production process included in the standard version of COSMO." Also, I assume the RS formulation uses the general 350 fragments per milligram rime value, but I would mention this value here.

*Yes, that is correct. We adapted the manuscript as follows:*

*A default value of 350 fragments per milligram of rime is used in the rime splintering parameterization.*

6. Lines 169-170 Could you say something more precise about what you mean by early graupel formation? As it stands, you say that it is "promoted when ice crystals or snow are converted to graupel" which is a bit of a tautology.

*We will clarify the meaning of early graupel formation with the following adaption to the manuscript:*

*In eq. (70) of Seifert and Beheng (2006), they specify that ice and snow crystals can only be converted to graupel once they reach $\bar{D}_{i,s} \geq 500 \, \mu m$. However, in the current version of the 2M scheme (as used in this study), ice and snow crystals are converted to graupel already once they exceed $\bar{D}_{i,s} \geq 200 \, \mu m$. Therefore, earlier graupel formation is promoted in the current version which should lead to enhanced SIP though ice-graupel collisions. To test the model's sensitivity to these different thresholds for graupel formation, we set-up sensitivity studies with graupel formation at $\bar{D}_{i,s} \geq 300, 400$ and $500 \, \mu m$, respectively, to understand how the conversion rate impacts SIP processes.*

7. Lines 190-191 "The vertical evolution of $K_{dp}$ and $Z_{DR}$ is similar, with a peak observed about 4 km amsl.." Here you are already looking at Figure 5, right? Please cite the figure.

*Correct. We adapted the reference.*

8. Line 208-209 Unless I am missing something, I would remove the sentence that IWC and NICE "fall within the 10 and $90^{th}$ percentiles range of the observations." This does not really indicate any agreement to me.

*We removed this entire paragraph because we discuss the limitations of the retrievals in the methods sections.*

9. Line 248-250 Given that no simulation performs best on all metrics, is it a fair conclusion that the size scaling from Sotiropoulou et al. (2020) (your Equation 3) is not an important factor in this case?

*No, this would not be a fair conclusion. In Eqs. 1 and 2 (now 4 and 5) a scaling parameter is applied to compensate for the discrepancy between ice particle size vs velocity used in Takahashi et al. (1995). Sotiropoulou et al. (2020) instead applied a direct size scaling in terms of the large ice particles that was used in Takahashi et al. (1995). We believe that the parameterization from Sotiropoulou et al. (2020) is most likely the better parameterization to use, but it needs to be tested in more cases. One caveat is in how the scaling factor is applied. The scaling factor has to be used with caution because very small graupel might not cause ice fractures in collisions with ice.*

10. Figure 6 Which simulation is this figure from?

*In Figure 6 (now Figure 8) we used the BR2.8T simulation. We do mention it in the text, however, we now added it to the figures (8, 10 and 11) to make it clearer. Thank you.*

11. Line 283 I would remove the "not surprisingly" here. There has been significant discussion of how updraft modulates SIP rates but not shear, so it is indeed surprising that longitudinal wind shear is the "most important determinant" here.

This is true, we removed "not surprisingly"

12. Line 349 I would write "Both shortcomings could be explained by omission of ice-graupel collisions." There are also other processes that could explain an overestimation of $Z_H$ and underestimation of NICE.

Thank you. We have adapted the manuscript.

13. Line 296-297 "The higher MI values for V-wind shear with SIP is most likely why the Wind shear had larger and significant MI values with SIP." I do not understand this; to me, it sounds like you are saying the values are larger because they are larger. Could you please reword or remove?

We removed this statement.

**References**

[revised manuscript text omitted]

---

## Author Comment (AC2)

**Heavy snowfall event over the Swiss Alps: Did wind shear impact secondary ice production?**

Zane Dedekind[*1], Jacopo Grazioli[*2], Philip H. Austin[1], and Ulrike Lohmann[3]

[1]Department of Earth, Ocean, and Atmospheric Sciences, University of British Columbia, Earth Sciences Building, 2207 Main Mall, Vancouver, BC, V6T 1Z4, Canada
[2]Environmental Remote Sensing Laboratory (LTE), École Polytechnique Fédérale de Lausanne (EPFL), Lausanne, Switzerland
[3]Institute of Atmospheric and Climate Science, ETH Zurich, Switzerland
[*]Equally contributing authors

We sincerely thank Reviewer 2 for the constructive feedback. The suggestions and comments considerably improved the quality of the manuscript.

Below we present a detailed response with the reviewer comments in black, our responses in blue and additions to the manuscript in blue italics.

**Specific comments**

1. It was difficult follow the entire manuscript, probably because of a lack of descriptions of model simulations (or I cannot find at least). The authors should give detailed, careful explanations of simulations for people who do not have model background. Specifically, I have the following questions:

   – For Eqs. 1-3, what does "BR" stand for? What do BR28, BR2.8T, and BR-Sot mean? Because I could not know them, understanding the following descriptions was very difficult for me.

   Indeed, we did not specifically mention what BR mean. BR refers to breakup. We also added which parameterization is associated with its reference:

   *In this study of the heavy snowfall event during which high $K_{dp}$ values were recorded, we use the parameterizations for ice-graupel collisional breakup (BR) from Dedekind et al. (2021, BR28 and BR2.8T) and Sotiropoulou et al. (2021, BR-Sot) in COSMO in different forms.*

   References to the simulations were given beneath the equations and their descriptions. However, we now provide some more context to the description.

   *Because of an inconsistency between the hail particles and their corresponding fall velocity used in Takahashi et al. (1995), which is described in more detail in Dedekind et al. (2021), all the parameterizations (Eqs. 4, 5 and 6) have scaling factors. Equations 4 and 5 were applied in Dedekind et al. (2021) for the BR28 and BR2.8T simulations*

*respectively. The BR28 simulation is scaled by $\alpha=10$ and has a slower decay rate of fragment number at warmer temperatures represented by $\gamma_{BR}=2.5$ and equation 5 is scaled by 100 while using the same decay rate of fragment numbers of $\gamma_{BR}=5$ as used in Sullivan et al. (2018) which was derived from Takahashi et al. (1995). Equation 6, for the BR-Sot simulation, was applied in Sotiropoulou et al. (2021). They used a scaling parameter, $\bar{D}/\bar{D}_0$, that was applied to the breakup parameterization from Sullivan et al. (2018) where $\bar{D}_0 = 0.02\,mm$.*

– What does N_BR mean?

$\aleph_{BR}$ is the number of fragments generated per collision given all the parameters in Eqs. 4, 5 and 6. We added this to the description of the equations.

– Lines 164-165: To me this sentence does not make sense at all. Need detailed explanations. What is scaled; what are BR, Sot, 2.8T?

Thank you, we added more detail to the description of the equations as described above.

– What is RS simulation? I could not easily find the description about the simulation.

We added more information to the text to make it clear.

*Simulations were conducted including several SIP processes, which consisted of ice-graupel collisions (as thoroughly discussed in section 2.2.2 below) and a control simulation, referred to as the rime splintering (RS) simulation, where only rime splintering was active as a SIP process.*

– It seems to me that the radar reflectivity from the simulations is very large. For many regions below 5-6 km, reflectivity attained or exceeded 30 dBZ for all simulations. The authors mentioned the size of simulated particles, but still I think too large for snow scattering at X-band, otherwise it was graupel. Was graupel produced in the entire cloud below 5 km? Please give detailed settings of calculation of reflectivity from the simulation data.

1) During the duration of the simulations, all simulations produced graupel, albeit at lower mass concentrations for the collisional breakup simulations (Fig. 5). There is a strong correlation between the graupel concentration and the reflectivity which is most likely the dominant factor in the overall calculation of reflectivity (Fig. 5a). 2) Rime splintering was very low, which is typical for wintertime MPCs (Fig. 7f here and e.g., Henneberg et al., 2017; Dedekind et al., 2021), the ice and snow crystals grew mainly by depositional growth and aggregation to much larger sizes compared to the collisional breakup simulations (Fig. 6c). The snow crystals, especially, were twice the size of those in the breakup simulations (Fig. 6b, over 2 mm in diameter). Below we show Hovmöller plots (Figs. 1 and 2) for the size diameters for ice crystals, snow, and graupel from 15:00 to 17:30 UTC showing that the snow and graupel particles reached diameters of over 5.1 mm and 8 mm, respectively. 3) The rain mass and number concentrations were also substantially higher in the rime splintering simulation. All these factors may have contributed to the large overestimation of $Z_H$ apart from the issues that arise from size sorting which is discussed in the manuscript. We added a similar description to Section 3.2.1.

[Figure]

**Figure 1.** Hovmöller diagrams of snow diameters for panels **(a)** RS , **(b)** BR28, **(c)** BR2.8T and **(d)** BR-Sot between 15:00 and 17:30 UTC.

Here we briefly show the calculation of the simulated $Z_{\mathrm{H}}$ (Appendix B, Seifert, 2002). Calculating $Z_{\mathrm{H}}$ from the two-moment cloud microphysics scheme would not be possible without approximations and assumptions. The following relationship for the radar reflectivity of drops ($Z_{\mathrm{w}}$), using the Rayleigh approximation for the cross-section of drops (Eq. 1), results in:

$$\eta_w = \frac{\pi^5 |K_w|^2}{\lambda_R^4} \int\limits_0^\infty D^6 f_w(D) dD \tag{1}$$

where $D$ is the particle diameter, $\lambda_R$ is the wavelength of radar radiation, $\eta_w$ is the volumetric liquid water content, $f_w(D)$ is the number density distribution function for liquid water and $K_w^2 = 0.93$ the dielectric constant of liquid water. The reflectivity factor for cloud water is given by:

$$\tilde{Z}_w = \frac{\lambda_R^4}{\pi^5 |K_w|^2} \eta_w = \int\limits_0^\infty D^6 f_w(D) dD = \left(\frac{6}{\pi \rho_w}\right)^2 \int\limits_0^\infty x^2 f_w(x) dx = \left(\frac{6}{\pi \rho_w}\right)^2 Z_w \tag{2}$$

[Figure]

**Figure 2.** The same as Figure 1, but for graupel.

where $\rho_w$ is the water density. Because of the backscatter behavior for the mass-equivalent diameter with regards to $\rho_w$ and $\rho_i$ (ice density), the same applies to graupel, which is described as a spherical ice particle

$$\eta_g = \frac{\pi^5 |K_i|^2}{\lambda_R^4} \left( \frac{6}{\pi \rho_i} \right)^2 \int\limits_0^\infty x^2 f_g(x) dx_1 \tag{3}$$

where $x$ is the particle mass, $f_g(x)$ is the number density distribution function for graupel and $K_i^2 = 0.176$ is the dielectric constant of ice. The radar reflectivity factor for ice particles (e.g., graupel) is given by:

$$\tilde{Z}_g = \frac{\lambda_R^4}{\pi^5 |K_w|^2} \eta_g = \frac{|K_i|^2}{|K_w|^2} \left( \frac{6}{\pi \rho_i} \right)^2 \int\limits_0^\infty x^2 f_g(x) dx = \frac{|K_i|^2}{|K_w|^2} \left( \frac{6}{\pi \rho_i} \right)^2 Z_g. \tag{4}$$

For melting ice particles, however, $K_w$ must be used instead of $K_i$. In our study, the surface and in-situ cloud temperatures were below $0\,°\mathrm{C}$. Therefore, more information on the reflectivity calculations for melting ice particles can be found in Seifert (Appendix B, 2002). Finally, the radar reflectivity factor is given by:

$$dBZ = \frac{10}{\ln 10} \ln \left[ \frac{Z_{radar}}{\mathrm{mm}^6\ \mathrm{m}^{-3}} \right] \tag{5}$$

where $Z_{radar}$ is the sum of the reflectivity calculated for each individual cloud particle category (e.g., cloud drops, raindrops, ice crystals, snow crystals, graupel and hail):

$$Z_{radar} = \left(\frac{6}{\pi\rho_w}\right)^2 \left[Z_c + Z_r + \frac{\rho_w^2}{\rho_i^2}\frac{|K_i^2|}{|K_w{}^2|}\left(Z_{ic} + Z_s + Z_g\right)\right]. \tag{6}$$

We added the description to Appendix A.

2. I also felt a lack of observational evidence of secondary ice production. The authors need to show observational data and explanations of the secondary ice production. Below are my comments.

– For the case, both KDP and ZDR coincidentally increased at the same altitude. A signature of large KDP and large ZDR does not necessarily represent secondary ice production. Rather, it can be interpreted as size growth of individual particles (without secondary ice production). One of good signatures of secondary ice production is large KDP collocated with small ZDR. This can be seen Fig. S3, but less description about this in the text. In addition to such signature, the previous literature also showed other observational evidence such as in-situ data, Doppler spectra, and/or liquid water path. This manuscript did not show such evidence.

We thank the reviewer for the comment and the suggestion about the interpretation of the polarimetric signatures. We believe that this question arose because we did not provide the Hovmöller diagrams for $K_{\mathrm{dp}}$ and $Z_{\mathrm{DR}}$ between 13:00 and 17:30 UTC, but only for individual time steps (in Fig. 6 and Fig. S3). Now, we include Figure 2 which is the Hovmöller plots for $K_{\mathrm{dp}}$ and $Z_{\mathrm{DR}}$ between 13:00 and 17:30 UTC, as done previously for $Z_{\mathrm{H}}$ and the hydrometeor classification. It is more evident now, as the reviewer noticed for Fig. S3, that the peak of $Z_{\mathrm{DR}}$ is mostly below the peak of $K_{\mathrm{dp}}$.

An interesting aspect of this case is that, while $Z_{\mathrm{DR}}$ reaches positive values that seldom exceed 1 dB (2 dB as reference for the highest values), $K_{\mathrm{dp}}$ reaches values above 1.5 / 2 $^\circ\mathrm{km}^{-1}$ more frequently, which is instead extremely large for observations collected during a snow event. Because $K_{\mathrm{dp}}$ is affected by the number concentration while $Z_{\mathrm{DR}}$ is not, we interpreted this as a clear signature of a very large number concentration of hydrometeors from the perspective of the radar data. We can interpret these radar signatures as SIP when they are put into context of the model simulation that included collisional breakup as a SIP mechanism (e.g., BR28, BR2.8T and BR-Sot).

Regarding the latter part of the reviewer comment, we believe that we have multi-source complementary evidence: the radar signatures, 2DVD observations and model simulations. Unfortunately, Doppler spectra were not available at this time. As for the 2DVD, the number concentration was among the highest ever recorded at this location for this instrument during its deployment period (2009 to 2011). The largest 10-minutes average value of the number

concentration measured by the 2DVD during this case corresponds roughly to the 99.9% quantile of all the observations collected by this instrument at this location. In this specific case, the particles are also small as illustrated in Figure 1 of this document (on average around 0.5 mm with the 99% quantiles below 2 mm). We have, therefore, a very large number of small particles in the population of hydrometeors. Here, the model provides the required complementary evidence of SIP by the collisional breakup processes: only the model simulations with breakup produce the same order of magnitude in terms of number concentration, and they represent the $Z_H$ profile better. We included the rime splintering simulation (RS) as our control simulation because the rime splintering process is traditionally used in numerical weather prediction models whereas collisional breakup is not. The RS simulation had very low to no SIP which resulted in less numerous but larger sized ice particles (Figs. 6a,b, 7f). These results do not agree with the 2DVD observations. In conclusion, we are confident that SIP occurred, not only from radar observations, but also by the concurrent information from the 2DVD and the collisional breakup simulations. The radar signatures, $K_{dp}$ in particular, are so intense that they require in-depth investigation on the nature of the ice produced.

[Figure]

**Figure 3.** Evolution of (10-min) number concentration, mean diameter and quantile 99% of the diameters of the hydrometeors measured by the ground-based 2DVD.

In the revised manuscript we further stress the magnitude of the $K_{dp}$ signature (Section *Case study*):

*In particular $K_{dp}$ reached values around 1.5 ° km$^{-1}$ at certain height levels and towards the end of the event it was exceeding 2 ° km$^{-1}$ (Fig. 2a). $K_{dp}$ scales with radar frequency. A statistical analysis of $K_{dp}$ in snowfall conducted with this radar and in this location but over a long observation period (Schneebeli et al., 2013), showed that the 80% quantile of $K_{dp}$ at every height level is lower than 0.5 ° km$^{-1}$. Considering that the distribution of $K_{dp}$ is very skewed, values above 1 ° km$^{-1}$ in snow can be considered as unusually large.*

In the Introduction we rephrased the explanation on the interpretation of $K_{dp}$ and $Z_{DR}$ with additional references to their interpretation in terms of SIP:

*. . . specific differential phase shift $K_{dp}$ [°$km^{-1}$]. This variable is complementary and not redundant: it is in fact not affected by the absolute calibration of a radar and is less affected than $Z_{DR}$ by eventual presence of large isotropic particles within the sampling volume. Local $K_{dp}$ enhancements in snowfall have been documented (Schneebeli et al., 2013; Bechini et al., 2013) and in some cases been associated with SIP (e.g., Andrić et al., 2013; Grazioli et al., 2015; Sinclair et al., 2016)*

We clarified also how the 2DVD was used in this study when introducing the instruments in Sec. 2:

*The 2DVD measures the size and fall velocity of hydrometeors larger than about 0.2 mm in terms of maximum dimension. The 2DVD is used in this study as ground reference to quantify the number concentration of snowfall particles (larger than 0.2 mm).*

– Because of less figures from the observations (there are only reflectivity and hydrometeor classification plots), it was difficult to follow the first and second paragraphs of Sect. 3.1.1.

This is true, we should have made reference to figures 6(d, e) and S3(d, e). Additionally, we now show also $K_{dp}$ and $Z_{DR}$ in the new figure 3. We have adapted the manuscript to refer to the appropriate figures. The first reviewer, rightfully, also had difficulty with the second paragraph of 3.1.1. We modified the text to be more concise:

*$Z_H$ was significantly overestimated by the RS simulation between 13:00 and 17:30 UTC which most likely was a result of the following chain of events. 1) Insufficient droplets of size 25 μm (Fig. 5d), within the narrow temperature range ($-3 \geq T \geq -8$ °C), led to a limitation in ice particle growth by riming and therefore limited rime splintering (Fig. 6f). 2) Because rime splintering was not that active, typical for wintertime MPCs (e.g., Henneberg et al., 2017; Dedekind et al., 2021), the ice and snow crystals grew mainly by depositional growth and aggregation. 3) The ice and snow crystal size distributions widened substantially (Figs. 6a, b and S2a, b). These categories both had number concentrations less than $100\,L^{-1}$ with particle diameters of up to 0.8 and 5.1 mm, respectively, at 15:30 UTC. 4) The larger ice and snow crystal diameters resulted in enhanced $Z_H$. These observation is consistent with other times during the day which showed even larger sized ice and snow crystals of 0.9 and 5.2 mm, respectively (Figs. S3a and S4a) as well as higher rain mass mixing ratios (e.g. Fig. 5a). There were single grid points where snow crystal even reached diameters of 13 to 17 mm during the latter part of the day (not shown here). Additionally, excessive size sorting in the model most likely contributed to the overestimation in $Z_H$. Size sorting typically occurs within the sedimentation parameterization of 2M schemes in regions of vertical wind shear or updraft cores (Milbrandt and McTaggart-Cowan, 2010; Kumjian and Ryzhkov, 2012). All these factors contributed to the RS simulation overestimating $Z_H$ by at least 8 dBz throughout the vertical profile compared to the observations (Fig. 7d).*

- RHI scans were performed from the low to high elevation angles (0-90 degrees). Observed Kdp and Zdr should have strong dependency on elevation angles. Did you correct the values for angles?

  While other studies used some correction for elevation (notably Ryzhkov et al., 2005), for this study we decided instead not to use elevation angles above $40°$ (this was not mentioned in the original manuscript, so we thank the reviewer). It must be noted that, as RHIs are summarized with statistics along the $x-$ dimension, most of the data correspond to elevation angles well below $40°$ of elevation. The reason behind the choice not to correct for the polarimatric variables is that, at the time of this case study, the radar was not performing a regular bird-bath calibration of $Z_{DR}$ and thus we cannot be confident that the accuracy of the differential reflectivity is around $0.1$ dB or better, which is a prerequisite when applying the correction in a meaningful way (see Fig. 12 of Ryzhkov et al., 2005). This is now clarified in the text as:

  *Only observations obtained at elevation angles below $40°$ are used in order to limit the effect of elevation dependencies on the polarimetric variables (Ryzhkov et al., 2005).*

- I was not sure how the 2DVD data were used other than number concentration. Did the 2DVD show good evidence of secondary ice production?

  Please see our response to major point #1 above.

- Please explain how to estimate NICE and IWC from observation (e.g. Fig. 6, Fig. S3).

  Following this comment and the suggestion of reviewer 1, we decided to include the full formulation of these retrievals as well as a thorough discussion of the inherent assumptions and shortcomings in Section 2.1. of the manuscript.

- Did you see shear instability?

  Thanks for this question. We calculated the bulk Richardson which is a ratio of the buoyant energy to shear-kinetic energy. The Richardson number determines the dynamic stability of the air layer which becomes turbulent if the Richardson number is less than the critical Richardson number of 0.25. In Fig. 2 we show where the air layer was dynamically unstable, mainly as a result of the wind shear. It supports our conclusion that the wind shear enhanced the turbulence. In this case, the turbulence caused higher SIP rates through ice-graupel collisions. We also now mentioned this in the text in Section 3.1.

  *Here, the bulk Richardson number, which is a ratio of the buoyant energy to shear-kinetic energy, is determined to asses the dynamic stability of the air layer. An air layer becomes turbulent if the Richardson number is less than the critical Richardson number of 0.25 (e.g., ?). Figure 2b-e show where the air layer was turbulent which enhanced the interactions between ice particles and could have caused enhanced ice-graupel collisions. Regions of enhanced updraft, where hydrometeors can grow to larger sizes, were mostly seen immediately above the turbulent layer.*

**Technical comments**

1. Three digits are needed for latitude/longitude of the instrument locations.

   We thank the reviewer and we provide now the 3 digits for the 2DVD and MXPOL.

2. Line 123 "dual-Doppler radar output": I was confused. Did you perform dual Doppler radar analysis (did you use two Doppler radars)? If so, please provide the second radar information.

   We thank the reviewer for spotting this confusion in the technical jargon. Indeed only one radar was used. We corrected the sentence to *dual-polarization*

3. Line 150: The use of consistent unit throughout the manuscript for temperature is better.

   We changed the temperature unit to °C.

4. 3.2: What criteria were used for separating the period?

   The three periods were seperated by the location and intensity of the wind shear and corresponding turbulent layer.

   13:00-14:15: Strong wind shear and turbulence mixing at lower altitudes below 3 km with a stronger updraft regions above the turbulent layer. Graupel beginning to form and grow in size.

   14:30-15:45: The turbulent layer ascends to 4 km and into temperature region where SIP from collisional breakup is more favored.

   16:00-17:15: The wind shear reduces substantially causing the turbulent layer to dissipate. The updraft in this period is reduced and the cloud is in a glaciated state which should not favor SIP.

   We added the following description to Section 3.3:

[revised manuscript text omitted]

---

## Referee Report (RR1)

**Review 2: Heavy snowfall event over the Swiss Alps: Did wind shear impact secondary ice production?**

I am grateful to the authors for putting effort into clarifying the manuscript, which I can see in their responses to my first round of comments. I will start by reiterating that I think modulation of secondary ice production rates by wind shear is an interesting idea and extracting such signals from remote sensing data is a worthy effort.

I have to admit, however, that I am still struggling to follow arguments in the results sections. I am wondering if you can still reorganize Section 3.2. Section 3.2.1 is entitled "Model and Doppler radar comparison." I would limit the discussion in this section to exactly that – comparisons of the modeled versus measured $Z_H$, $K_{dp}$, and $Z_{DR}$ values – and focus on Figures 3, 4, and what is currently Figure 7d and e. I would extract 7d and 7e into a new, separate figure. Thereafter, perhaps Section 3.2.2 can be "Modeled and measured cloud properties" in which you focus on Figure 7a,b,c, and f *and* Section 3.2.3 "Microphysical explanations" in which you present some potential pathways for the model-measurement differences on the basis of Figures 5 and 6 (e.g. Lines 274-285 could be migrated to such a section.)

Sections 3.3 and 3.4 are somewhat easier to follow, although it seems to me that a better title for Section 3.3 would be something like *Linking Simulated Large-Scale Wind and SIP*. I would also work to end each of the subsections within Section 3 with a "bottom line," i.e. the one or two points you want the reader to take away from the discussion there. This would help readability.

**(Relatively More) Major Comments**

**Abstract** – Can you start with a more general sentence in the abstract? What are you trying to understand or elucidate in this study?

**Lines 65-85** – I appreciate that these conceptual definitions of the remote sensing variables have been added; indeed, I requested that during the first round. However, placed here, it interrupts the flow of the study motivation and approach, and I think it belongs better in Section 2.1.

**Lines 153-155** – "The model output was interpolated along the mean of three vertical cross-sectional paths, ≈ 3 km wide … The cross-sections of the simulations were averaged to create a mean cross-section." I was a bit confused about this during the first read-through also. Figure 1 shows only two such cross-section paths, as far as I can see. How was a separation of 3 km between these paths chosen? And it seems like one of the cross-sections cuts more or less directly through the Doppler radar location; is that right? I am also curious about the variability between the three cross sections. I guess this is as fine a method as any to compare to radar data, but I would appreciate another sentence or two explaining its setup and uncertainty.

**Lines 183-185** – "During collisional breakup graupel collides with either ice and/or snow particles and fractures." I would start here with a more general definition of collisional breakup. A version between ice and graupel is being employed here, but it refers more generally to the collision of any two frozen hydrometeors in which the density difference (and hence terminal velocity difference) is sufficient that the collisional impact causes shattering.

**Lines 185- 192** – I would refocus / remove these sentences to make two points. First, almost no empirical constraints exist for the efficiency of any form of collisional breakup, and second, from a theoretical standpoint, collisional kinetic energy between the hydrometeors is the key parameter for this efficiency. These points are more fundamental than discussions of ice-graupel, ice-snow, etc. because these distinctions are artificial and will disappear as we transition toward particle properties schemes.

**Line 211** – This has become quite a long paragraph. I would break after "from Sullivan et al. (2018) where $D_0 = 0.02m$" and add a transition sentence that indicates that *Previous studies indicate that artificial thresholds for hydrometeor numbers or conversion rates may strongly influence the output of secondary ice production parameterization.*

**Line 230** – As you introduce the case study here, I think it is important to mention some time scales already. You will be looking at about 4 hours of front evolution with high-intensity snowfall concentrated over 1.5 hours during that time.

**Lines 289-290** – Is this a single, time-averaged observation of ICNC from the disdrometer?

**Lines 345-346 *and* Lines 349** – I have some reservations here about your wording in regard to the role of V-wind shear. Covariability does not necessarily imply that a factor is a "determinant" or that it "played a major role." I might limit the statement to something like "Environment of high meridional wind shear tend to be the same environments of high secondary ice production rates."

**(Relatively More) Minor Comments**

**Line 2** – "high precipitation" – *high intensity precipitation*?

**Lines 33-35** – "The enhancement of smaller ice particles triggers an increase in the combined growth rates (riming and deposition) of up to 33% resulting in larger latent heat release… When ice-ice collisions occur in wintertime orographic MPCs, the general tendency is for riming to decrease." To me, there are two contradictory statements being made here. Do existing results indicate that riming rates decrease or increase in the presence of secondary ice production? Or is this dependent on the fragment sizes being generated?

**Line 53** – new paragraph after "once they reach a size of 200 um"? You are transitioning to discuss remote sensing technique to study these processes.

**Line 57** – "Two non mutually exclusive approaches can be found in the literature." I appreciate that the authors have added more literature review regarding use of remote sensing to infer secondary ice production. It was not clear to me why the two approaches were not mutually exclusive?

**Line 63** – "Additional information (in-situ, models, or a combination of more radars)…" in-situ *data*

**Line 66** – "The waves interact with precipitation" But also with suspended condensate, right?

**Line 75** – new paragraph after "as is the propagation speed of the waves"? Then you separate discussion of reflectivities from that of differential phase shift.

**Line 170** – "The primary production of ice formation is described by.." *Primary ice production occurs via..*

**Lines 178-179** – "Secondary ice production through rime splintering is the only process that is included in the standard version of COSMO which has been used extensively in other numerical weather models" I think you are trying to say that rime splintering implementations are fairly widespread in other numerical weather models than COSMO, but the wording is confusing. I would remove *which has been used extensively in other numerical weather models.*

**Lines 202-204** – "Only using ice-graupel collisions would limit the full description of SIP as a result of wind shear when graupel formation becomes restricted." I do not understand what this means. Can you clarify somehow?

**Lines 224** – "For this purpose a 10 km x 10 km region was selected and masked by the levels in which SIP occurred (T > -21C, *blue box in Fig. 1)*"

**Figure 2 caption** – "The blue and red colors denote wind blowing towards and away from the radar" I don't see blue and red? I'm understanding the blue-brown shading to be horizontal wind relative to the mean cross-section and the blue-brown contours to be vertical wind. I am also not really extracting "the shaded gray area" which denotes cloud area fraction.

**Line 241** – (Figs. 2*c*-e) Fig. 2b is not a collisional breakup simulation.

**Line 243** – Missing reference

**Line 274** – Missing reference

---

## Author Response (AR2)

**Heavy snowfall event over the Swiss Alps: Did wind shear impact secondary ice production?**

Zane Dedekind[*1], Jacopo Grazioli[*2], Philip H. Austin[1], and Ulrike Lohmann[3]

[1]Department of Earth, Ocean, and Atmospheric Sciences, University of British Columbia, Earth Sciences Building, 2207 Main Mall, Vancouver, BC, V6T 1Z4, Canada
[2]Environmental Remote Sensing Laboratory (LTE), École Polytechnique Fédérale de Lausanne (EPFL), Lausanne, Switzerland
[3]Institute of Atmospheric and Climate Science, ETH Zurich, Switzerland
[*]Equally contributing authors

We thank both reviewers for the constructive feedback. The additional suggestions and comments improved the readability of the manuscript.

Below we present a detailed response with the reviewer comments in black, our responses in blue and additions to the manuscript in blue italics.

**Reviewer 1: Major comments**

1. Abstract: Can you start with a more general sentence in the abstract? What are you trying to understand or elucidate in this study?

   We agree and have added the following few sentences for a more general audience.

   *The change in wind direction and speed with height, referred to as vertical wind shear, causes enhanced turbulence in the atmosphere. As a result, there are enhanced interactions between ice particles that break up during collisions in clouds which could cause heavy snowfall. For example,*

2. Lines 65-85 – I appreciate that these conceptual definitions of the remote sensing variables have been added; indeed, I requested that during the first round. However, placed here, it interrupts the flow of the study motivation and approach, and I think it belongs better in Section 2.1.

   We moved the section to Section 2.1.

3. Lines 153-155 – "The model output was interpolated along the mean of three vertical cross-sectional paths, $\approx$ 3 km wide ... The cross-sections of the simulations were averaged to create a mean cross-section." I was a bit confused about this during the first read-through also. Figure 1 shows only two such cross-section paths, as far as I can see. How was

[Figure]

**Figure 1.** Hovmöller diagrams of the horizontal reflectivity ($Z_H$) from the Doppler radar for the BR2.8T simulation between 13:00 and 17:30 UTC. The variability between the three cross-sections (CS) is illustrated in the panels for CS1, CS2 and CS3. Note that the color bar is different from the figures used in the manuscript to better show differences between the CSs of the BT2.8T simulation.

a separation of 3 km between these paths chosen? And it seems like one of the cross-sections cuts more or less directly through the Doppler radar location; is that right? I am also curious about the variability between the three cross-sections. I guess this is as fine a method as any to compare to radar data, but I would appreciate another sentence or two explaining its setup and uncertainty.

The cross-sections shown in Fig. 1 are the outer limits of the cross-sections. Not shown in Fig. 1 is another cross-section which is between the outer cross-sections. The separation between each cross-section is about 1 km which is similar to the model resolution of 1 km. Yes, one of the cross-sections cuts more or less directly through the Doppler radar location. The variability between the cross-sections is small in this case study (Fig. 1), but a necessary step especially when comparing observations to model simulations in mountainous regions. The section was modified to include more setup details.

*To account for the uncertainty associated with simulating atmospheric processes in mountainous terrain, three cross-sections were interpolated from the model output (only the outer two cross-sections are shown in Fig. 1). Each cross-section, of which one cross-section cuts more or less through the location of the radar, is separated by 1 km which is similar to the model resolution. The direction of each of the cross-sections is similar to the direction of the generated RHI cross-section from the weather radar. The three cross-sections are then averaged and compared to the radar data.*

4. Lines 183-185 – "During collisional breakup graupel collides with either ice and/or snow particles and fractures." I would start here with a more general definition of collisional breakup. A version between ice and graupel is being employed here, but it refers more generally to the collision of any two frozen hydrometeors in which the density difference (and hence terminal velocity difference) is sufficient that the collisional impact causes shattering.

Thank you for the suggestion. We have added the following text to the manuscript.

*Generally, collisional breakup refers to the collision of any two frozen hydrometeors with different densities in which the collisional kinetic energy is sufficient that the collisional impact causes shattering. Here, collisional breakup is when either ice and/or snow particles collide with graupel and fracture.*

5. Lines 185- 192 – I would refocus / remove these sentences to make two points. First, almost no empirical constraints exist for the efficiency of any form of collisional breakup, and second, from a theoretical standpoint, collisional kinetic energy between the hydrometeors is the key parameter for this efficiency. These points are more fundamental than discussions of ice-graupel, ice-snow, etc. because these distinctions are artificial and will disappear as we transition toward particle properties schemes.

This is true. We modified this section with the following:

*Almost no empirical constraint, apart from Takahashi et al. (1995) who used collisions between hail-sized particles, exists for the efficiency of any form of collisional breakup. The collisional kinetic energy and the density between hydrometeors are important parameters for this efficiency.*

6. Line 211 – This has become quite a long paragraph. I would break after "from Sullivan et al. (2018) where D0 = 0.02m" and add a transition sentence that indicates that "Previous studies indicate that artificial thresholds for hydrometeor numbers or conversion rates may strongly influence the output of secondary ice production parameterization."

We included the break in the paragraph.

7. Line 230 – As you introduce the case study here, I think it is important to mention some time scales already. You will be looking at about 4 hours of front evolution with high-intensity snowfall concentrated over 1.5 hours during that time.

It is a good suggestion and we included the timescale in the modified section below.

*A synoptic system passed over Switzerland on 26 March 2010 during which we analyze the evolution of a cold front from 13:00 to 17:30 UTC with intense precipitation from 15:00 to 17:30 UTC. Furthermore, the cold front was associated with a surface temperature drop of $\sim 7\,^\circ C$ (Fig. S1), a south-westerly wind flow at higher altitudes, vertical wind shear closer to the surface below 4 km amsl and the development of peculiar polarimetric radar signatures (e.g. an intense $K_{\mathrm{dp}}$).*

8. Lines 289-290 – Is this a single, time-averaged observation of ICNC from the disdrometer?

The green triangle in the figure referenced here indicates the total volumic number concentration from the 2DVD computed at a time scale of 5 minutes (meaning that an observation at 15:30 UTC is representative of data between 15:25 and 15:30 UTC). The temporal resolution is now explicitly mentioned in the text.

9. Lines 345-346 and Lines 349 – I have some reservations here about your wording in regard to the role of V-wind shear. Covariability does not necessarily imply that a factor is a "determinant" or that it "played a major role." I might limit the statement to something like "Environment of high meridional wind shear tend to be the same environments of high secondary ice production rates."

We agree with the reservation and have modified the manuscript.

**Reviewer 1: Minor comments**

1. Line 2 – "high precipitation" – high intensity precipitation?

We modified the text to say high-intensity precipitation.

2. Lines 33-35 – "The enhancement of smaller ice particles triggers an increase in the combined growth rates (riming and deposition) of up to 33% resulting in larger latent heat release... When ice-ice collisions occur in wintertime orographic MPCs, the general tendency is for riming to decrease." To me, there are two contradictory statements being made here. Do existing results indicate that riming rates decrease or increase in the presence of secondary ice production? Or is this dependent on the fragment sizes being generated?

It appears to be contradictory, however, Dedekind et al. (2021) show that the increased combined growth rates of ice particles include a strong reduction in the riming rate overshadowed by an even stronger enhancement in depositional growth rate. The results indicate that riming decreases in the presence of secondary ice production because the fragmented ice particles are smaller limiting their interaction with supercooled liquid water. We adapted the text to make this clearer.

3. Line 53 – new paragraph after "once they reach a size of 200 um"? You are transitioning to discuss remote sensing technique to study these processes.

Yes, we started a new paragraph.

4. Line 57 – "Two non mutually exclusive approaches can be found in the literature." I appreciate that the authors have added more literature review regarding use of remote sensing to infer secondary ice production. It was not clear to me

why the two approaches were not mutually exclusive?

The approaches are not mutually exclusive meaning that some radar systems have the capability to do both. We rephrased this sentence to remove the phrasing *non mutually exclusive*, as we agree that it is possibly confusing for the reader.

5. Line 63 – "Additional information (in-situ, models, or a combination of more radars)..." in-situ data
   Yes.

6. Line 66 – "The waves interact with precipitation" But also with suspended condensate, right?
   That is correct, we added it.

7. Line 75 – new paragraph after "as is the propagation speed of the waves"? Then you separate discussion of reflectivities from that of differential phase shift.
   A new paragraph was started.

8. Line 170 – "The primary production of ice formation is described by.." Primary ice production occurs via..
   We modified the text accordingly.

9. Lines 178-179 – "Secondary ice production through rime splintering is the only process that is included in the standard version of COSMO which has been used extensively in other numerical weather models" I think you are trying to say that rime splintering implementations are fairly widespread in other numerical weather models than COSMO, but the wording is confusing. I would remove which has been used extensively in other numerical weather models.
   Correct, we modified the text to:
   *Secondary ice production through rime splintering, which is widely used in numerical weather prediction models, is the only process...*

10. Lines 202-204 – "Only using ice-graupel collisions would limit the full description of SIP as a result of wind shear when graupel formation becomes restricted." I do not understand what this means. Can you clarify somehow?
    Yes. Only using ice-graupel collisions, and not ice-snow and ice-ice collisions, means that SIP might be underestimated in an environment of strong wind shear. However, we removed this sentence because it creates confusion within its context.

11. Lines 224 – "For this purpose a 10 km x 10 km region was selected and masked by the levels in which SIP occurred (T > -21C, blue box in Fig. 1)"

The additional information was added.

12. Figure 2 caption – "The blue and red colors denote wind blowing towards and away from the radar" I don't see blue and red? I'm understanding the blue-brown shading to be horizontal wind relative to the mean cross-section and the blue-brown contours to be vertical wind. I am also not really extracting "the shaded gray area" which denotes cloud area fraction.

We updated the caption and removed the cloud area fraction from Fig. 2.

13. Line 241 – (Figs. 2**c**-e) Fig. 2b is not a collisional breakup simulation.

We corrected the reference.

14. Line 243 – Missing reference

The link to the reference was corrected.

15. Line 274 – Missing reference

The missing reference was updated as well as other incorrect references to figures in this section.

**Reviewer 2: Minor comments**

1. I agree with the authors response to my comment 2, however, it is still unclear to me how evidence of SIP was observed. As I commented in the previous review, SIP might be explained as high Kdp collocated with low Zdr, which could be explained by large number concentration of oblate, smaller ice particles (high Kdp) and near-spherical, large snow (low Zdr). This could be observed between 16:21 and 17:21 UTC (Fig.3). However, Fig. 7e showed peak of Zdr is consistent with the peak of Kdp. This could represent slightly-oblate large ice particles, rather than different shape/size of particles.

SIP can be generated by various collisional breakup mechanisms, possibly generating different radar signatures. For example, at around 15:30 UTC collisional breakup was driven by the presence of graupel as opposed to collisional breakup driven by fragile ice aggregates which occurred later during the day. The collisional breakup parameterizations used in this study only included ice-graupel collisions (collisions of particles of different densities). Therefore, the simulations are only able to replicate SIP during the early phase of the event when graupel was present. For this reason, we focussed on the earlier phase of the event. We agree with the reviewer that the second phase of the event more closely resembles the expected SIP signature (high Kdp, low Zdr). At the same time, we believe that this is associated with the breakup

of fragile hydrometeors in the absence of graupel, a mechanism that the model is not able to replicate which should definitely be addressed in future research and compared with these prominent radar signatures (especially $K_{dp}$). In the conclusion section, this is discussed as follows:

*Another time period with high $K_{dp}$, but low $Z_{DR}$ was observed at 17:00 UTC which was not captured by the breakup simulations because the graupel mixing ratio was depleted. The breakup parameterization does not include ice-ice collisions and relies only on graupel as the collider specie. At this time the radar signatures suggested that collisions of dendrites caused the formation of small oblate particles (increasing $K_{dp}$) but also the formation of a few, larger, isotropic aggregates (decreasing $Z_{DR}$).*

The presence of dendrites was highly likely given the favorable temperature range (discussed earlier in the manuscript). At 15:30, the model shows that ice-graupel collision is the dominant mechanism. The radar-based hydrometeor classification suggests the presence of rimed hydrometeors, corroborating this hypothesis, and the 2DVD shows a high number concentration and the absence of a significant number of big particles (see also the PSD spectra later in this document, so we can rule out the possibility of "slightly-oblate large particles" dominating the radar signal). Therefore, in an environment where graupel-ice collision SIP is taking place, we can have the presence of a large number of oblate, possibly rimed, particles (increasing $K_{dp}$), co-located with positive $Z_{DR}$ if the largest particles (rimed particles and graupel) are oblate. Also, it should be noted that $Z_{DR}$, at this time step, is positive but not extremely large (around 1 / 1.5 dB). Populations of rimed particles with signatures in this range are documented for example in Grazioli et al. (2015); Besic et al. (2016). In the revised manuscript, while describing the radar measurements (Sec. 3.2.1), we rephrased this as follows:

*The vertical evolution of $K_{dp}$ and $Z_{DR}$ is similar, with a peak observed about 4 km amsl, which is 1 km above the peak in $Z_H$ (Fig. 7d). The large and colocated values of $Z_{DR}$ and $K_{dp}$ suggest that a large population of oblate and rimed particles, without a significant presence of large isotropic hydrometeors, were present.*

2. Why did you select that time (15:30) for Fig.7e?

As mentioned above, this time step was selected because of the presence of graupel/rimed particles (both in the radar hydrometeor classification and in the model) that could lead to significant breakup-driven SIP. At later time steps, though interesting to study for the strong radar signatures, graupel was depleted and therefore the model, which requires graupel to initiate collisional SIP, cannot reproduce the observations. This discrepancy is in itself a result of this work and highlights the current limitations of the model. Therefore, it was discussed in the conclusions (see also the previous answer).

3. Also, I am wondering if 2DVD observed particle size distributions that represented SIP; for example high number concentration of smaller particles coexisting with large particles compared to non-SIP periods.

We must be cautious in the interpretation of 2DVD PSD spectra in terms of SIP, given the resolution of the instrument (0.2 mm, which cuts out the most interesting part of the distribution of small particles) and the absence of significant literature/methods for SIP detection using this imager. This is the main reason why we only used the 2DVD to retrieve

hydrometeor number concentrations at the ground level. This said it is indeed interesting to look at the spectra of this case study (see Fig. 2 of this document). We compare here a size spectrum collected at 1530 UTC with one collected at 1720 UTC. Both show a very large number of small particles (and we must keep in mind that particles smaller than 0.2 mm are not measured) but also larger particles of a few mm. The spectra collected at 1720 UTC has more large particles, which is compatible with the hypothesized presence of aggregates of dendrites in this time interval. Given that the amount of information that we can extract from the spectra is relatively minor regarding SIP, we decided not to include this type of data in the manuscript.

[Figure]

**Figure 2.** Two spectra of distribution of particle size measured by the 2DVD near the ground at two different time steps, each collected over 10 minutes of measurements. Note that this is a histogram of particle sizes crossing the measurement area in the given time and not strictly speaking a volumic PSD.

4. I do not still know what the numbers mean for the simulation names, i.e. BR28 and BR2.8T. This may be explained in Dedekind et al. (2021), but if it would be also explained in the current text, it would be helpful.

Table 1 was included in the text which describes the simulations.

**Table 1.** Sensitivity settings for the collisional breakup (BR) parameterization. The conversion rate (conv) is the size, in μm, at which rimed ice or snow are converted to graupel, $\alpha$ is the scale factor, $F_{BR}$ the fragments generated and $\gamma_{BR}$ the decay rate of fragment number at warmer temperatures. When $\gamma_{BR} = 5$, as used in the Takahashi parameterization, then T is included in the simulation name.

| conv | $\alpha$ | $F_{BR}/\alpha$ | $\gamma_{BR} = 5$ | $\gamma_{BR} = 2.5$ |
|------|----------|-----------------|-------------------|---------------------|
| 200  | 1        | 280             | BR-Sot            |                     |
| 200  | 10       | 28              |                   | BR28                |
| 200  | 100      | 2.8             | BR2.8T            |                     |
| 300  | 100      | 2.8             | BR2.8T_300        |                     |
| 400  | 100      | 2.8             | BR2.8T_400        |                     |
| 500  | 100      | 2.8             | BR2.8T_500        |                     |

**References**

Besic, N., Figueras i Ventura, J., Grazioli, J., Gabella, M., Germann, U., and Berne, A.: Hydrometeor classification through statistical clustering of polarimetric radar measurements: a semi-supervised approach, Atmospheric Measurement Techniques, 9, 4425–4445, https://doi.org/10.5194/amt-9-4425-2016, publisher: Copernicus GmbH, 2016.

Dedekind, Z., Lauber, A., Ferrachat, S., and Lohmann, U.: Sensitivity of precipitation formation to secondary ice production in winter orographic mixed-phase clouds, Atmospheric Chemistry and Physics, 21, 15 115–15 134, https://doi.org/10.5194/acp-21-15115-2021, 2021.

Grazioli, J., Tuia, D., and Berne, A.: Hydrometeor classification from polarimetric radar measurements: a clustering approach, Atmospheric Measurement Techniques, 8, 149–170, https://doi.org/10.5194/amt-8-149-2015, publisher: Copernicus GmbH, 2015.

Takahashi, T., Nagao, Y., and Kushiyama, Y.: Possible High Ice Particle Production during Graupel–Graupel Collisions, Journal of the Atmospheric Sciences, 52, 4523–4527, https://doi.org/10.1175/1520-0469(1995)052<4523:PHIPPD>2.0.CO;2, 1995.